



# Pyrite oxidization accelerates bacterial carbon sequestration in copper mine tailings Type of contribution

Yang Li[1,2], Zhaojun Wu[1], Xingchen Dong[3], Zifu Xu[1], Zhongjun Jia[2], Qingye Sun*,[1]

[1]School of Resources and Environmental Engineering, Anhui University, Hefei, Anhui Province, China
[2]State Key Laboratory of Soil and Sustainable Agriculture, Institute of Soil Science, Chinese Academy of Sciences, Nanjing, Jiangsu Province, China
[3]College of Resources and Environmental Sciences, Gansu Agricultural University, Lanzhou, Gansu Province, China

*Correspondence to*: Qingye Sun (sunqingye@ahu.edu.cn)

**Abstract.** Polymetallic mine tailings have great potential as carbon sequestration tools to stabilize atmospheric $CO_2$ concentrations. However, previous studies focused on carbonate mineral precipitation, while the role of autotrophs in carbon sequestration by mine tailings has been neglected. In this study, carbon sequestration in two mine tailings treated with $FeS_2$ and $^{13}C$-labeled $CO_2$ was analyzed using $^{13}C$ isotope labeling, pyrosequencing and DNA-based stable isotope probing (SIP) to identify carbon fixers. Mine tailings treated with $FeS_2$ exhibited a higher percentage of $^{13}C$ atoms ($1.76\pm0.06$ in Yangshanchong and $1.36\pm0.01$ in Shuimuchong) than the control groups over a 14-day incubation, as well an increase in the total organic carbon (TOC) content ($0.20\pm0.11$ mg/g in Yangshanchong and $0.28\pm0.14$ mg/g in Shuimuchong). These data demonstrated the role of autotrophs in carbon sequestration with pyrite addition. Pyrite treatment led to changes in the composition of bacterial communities, and the genera *Sulfobacillus* (8.04%) and *Novosphingobium* (8.60%) were found to be dominant in these communities. In addition, the DNA-SIP results indicated that the *cbbL* gene copy number was 8.20-16.50 times greater than the *cbbM* gene copy number in $^{13}C$-labeled heavy fractions. Furthermore, a *Sulfobacillus*-like *cbbL* gene sequence (cbbL-OTU1) accounted for 30.11-34.74% of all *cbbL* gene sequences in the $^{13}C$-labeled heavy fractions of mine tailings treated with $FeS_2$. These findings highlight the importance of the RubisCO form I-encoding gene, *cbbL*, in bacterial carbon sequestration and demonstrate the ability of chemoautotrophs to sequester carbon during sulfide mineral oxidation in mine tailings. This study is the first to investigate carbon sequestration by autotrophic groups in mine tailings through the use of isotope tracers and DNA-SIP.

**Keywords:** mine tailings; pyrite oxidation; autotrophic bacteria

## 1. Introduction

Carbon sequestration has been considered the leading strategy to stabilize $CO_2$ concentrations (Lal, 2004). Polymetallic mine tailings have great potential as a carbon sink for carbon sequestration because atmospheric $CO_2$ can be sequestered through the carbonation of noncarbonate minerals in mine tailings, including the dissolution of silicates, hydroxides and oxides and the precipitation of carbonate minerals (McCutcheon et al., 2016;Meyer et al.,



2014;McCutcheon et al., 2014). The strategy to sequester $CO_2$ as carbonate mineral is environmentally benign and
stable (Matter Juerg et al., 2007); however, sequestered carbon is also present in the form of inorganic carbon.

In terrestrial ecosystems, atmospheric $CO_2$ can be fixed into plants by photosynthesis and assimilated into soils as
soil organic carbon (SOC) by decomposition and microbial activity (Deng et al., 2016;Antonelli et al., 2018). However,
polymetallic mine tailings exhibit specific features, including a lack of organic matter, nutrients and nutrient-holding
capacity (Lottermoser, 2010;Young et al., 2015), that restrict plant growth, and plant productivity is generally difficult
to restore in mining wastelands (Hu et al., 2018;Li et al., 2017). Therefore, microbes may be the main source of
organic carbon in mine tailings. The limited amount of organic matter in mine tailings also inhibits the activities of
heterotrophic microorganisms, and therefore the microorganisms in these environments are lithotroph-dominant (Li et
al., 2015).

Polymetallic mine tailings contain sulfide minerals (e.g., pyrite), and the oxidation of these sulfide minerals leads to
a decrease in pH, also known as mine tailing acidification. With decreased pH, the relative abundances of the
Euryarchaeota and Actinobacteria phyla were found to significantly increase (Chen et al., 2013;Liu et al., 2014).
Euryarchaeota compete with β-Proteobacteria for ecological niches under such acidic conditions (Liu et al., 2014).
Actinobacteria utilize different types of resistance mechanisms to deal with heavy metals (Osanai et al., 2015), and this
phylum was found to be the main group participating in carbon and nitrogen biogeochemical processes in acidic mine
tailings. In parallel, acidophilic, chemoautotrophic bacteria, including *Acidithiobacillus*, *Leptospirillum* and
*Sulfobacillus* (Chen et al., 2013;Liu et al., 2014), can promptly participate in ferrous and sulfur oxidation in mine
tailings, and these autotrophic taxa play leading roles in carbon cycling and energy flow during the mine tailing
acidification process. Nevertheless, the relationship between the oxidation of sulfide minerals and carbon sequestration
by these acidophilic, chemoautotrophic bacteria is still unknown.

Currently, chemolithoautotrophic organisms fix atmospheric $CO_2$ by six pathways, including the widely distributed
Calvin-Benson-Bassham (CBB) cycle, the reductive tricarboxylic acid (rTCA) cycle, the reductive acetyl-CoA
pathway and the recently discovered 3-hydroxypropionate/4-hydroxybutyrate (HP/HB) cycles (Berg, 2011;Alfreider et
al., 2017). The ribulose-1,5-bisphosphate carboxylase/oxygenase (RubisCO) enzyme is the most prominent enzyme on
earth (Raven, 2013) and is important in the CBB cycle. In addition, the genes encoding the large subunit of RubisCO
serve as autotroph markers (Alfreider et al., 2017;Berg, 2011). The *cbbL* and *cbbM* genes encode RubisCO form I and
form II, respectively, with 25 to 30% amino acid sequence identity (Tabita et al., 2008).

In the present study, we conducted a microcosm experiment using mine tailings collected from two mines to
determine the effects of sulfide mineral (pyrite) oxidation on carbon sequestration in mine tailings through the addition
of pyrite. In addition, the main carbon fixers were analyzed through DNA-based stable isotope probing (DNA-SIP) and
*cbbL* and *cbbM* gene analysis. Our objectives were to investigate whether sulfide mineral oxidation can stimulate





carbon sequestration in mine tailings and to identify the key carbon sequestration groups in mine tailings during the acidification process.

## 2. Materials and Methods

### 2.1 Sampling of mine tailings

Samples of mine tailings were collected from the Tongling Yangshanchong (30°54′N, 117°53′E) and Shuimuchong (30°55′N, 117°50′E) mine tailing ponds of copper mines in Anhui Province, East China. Samples of oxidized mine tailings were collected on the surface using a steel corer in October 2015. The properties of the mine tailings were as follows: Yangshanchong acidic samples, pH 3.21, the total nitrogen (TN) 0.11 g·kg$^{-1}$, the total organic carbon (TOC) 16 g·kg$^{-1}$, $SO_4^{2-}$ 13.32 g·kg$^{-1}$, $As_T$ 63.29 mg·kg$^{-1}$, $Fe_T$ 133.46 g·kg$^{-1}$, $Cu_T$ 1.95 g·kg$^{-1}$, $Pb_T$ 27.58 mg·kg$^{-1}$, and $Zn_T$ 205.44 mg·kg$^{-1}$; Shuimuchong acidic samples, pH 2.92, TN 0.11 g·kg$^{-1}$, TOC 18 g·kg$^{-1}$, $SO_4^{2-}$ 8.84 g·kg$^{-1}$, $As_T$ 51.77 mg·kg$^{-1}$, $Fe_T$ 117.59 g·kg$^{-1}$, $Cu_T$ 2.53 g·kg$^{-1}$, $Pb_T$ 30.43 mg·kg$^{-1}$, and $Zn_T$ 176.59 mg·kg$^{-1}$.

### 2.2 DNA-SIP microcosms

Microcosms were constructed in triplicate for two mine tailing samples. The microcosms were incubated with 10% $^{13}$C-$CO_2$ as a labeling treatment, and the control microcosms were incubated with 10% $^{12}$C-$CO_2$. For each treatment, fresh mine tailings (equivalent to 10.0 g d.w.s.) were mixed with a total of 2 g of sterile pulverized $FeS_2$ at approximately 60% maximum water-holding capacity, followed by incubation at 25 °C in the dark for 14 days. Yangshanchong mine tailing samples (YM) cultured with $FeS_2$ are abbreviated as YM_$FeS_2$, and Shuimuchong mine tailing samples (SM) cultured with $FeS_2$ are abbreviated as SM_$FeS_2$. Fresh mine tailings at approximately 60% maximum water-holding capacity without any additive were used as the control groups and abbreviated as YM_ck and SM_ck.

### 2.3 Chemical properties analysis

Carbon isotope composition was analyzed by a Delta V Advantage Mass Spectrometer (Thermo Fisher Scientific, Inc., USA) coupled with an elemental analyzer (Flash2000; HT Instruments, Inc., USA) in continuous flow mode. The $^{13}$C atom % was calculated as follows:

$$^{13}\text{C atom \%} = \frac{[^{13}\text{C}]}{[^{13}\text{C}]+[^{12}\text{C}]} \times 100$$

The total organic carbon content was analyzed after acidification pretreatment using an element analyzer (Vario MACRO cube, Elementar INC., Germany), as described previously (Wang et al., 2015). The pH of mine tailing samples was measured using a pH meter (tailings:water=1 g:5 mL) at the end of the microcosm experiment. The $Fe^{2+}$





and $Fe^{3+}$ content were measured by chemical extraction techniques, as described previously (Heron et al., 1994). The total sulfate ion content was determined via ion chromatography after extraction by sodium hydroxide, as described

previously (Yin and Catalan, 2003).

## 2.4 DNA extraction and SIP gradient fractionation

Total DNA was extracted from each sample with the FastDNA® SPIN Kit for Soil (MP Biomedicals, Cleveland, OH, USA) according to the manufacturer's instructions. DNA-based stable isotope probing (DNA-SIP) fractionation was

performed as previously described (Zheng et al., 2014), and a total of 14 gradient fractions were generated for each sample. The refractive index of each fractionated DNA was measured via an AR200 digital hand-held refractometer (Reichert, Inc., Buffalo, NY, USA).

## 2.5 Real-time quantitative PCR analysis of the *cbbL* and *cbbM* genes

Real-time quantitative PCR analysis of the 16S rRNA gene was performed on a CFX96 optical real-time detection system (Bio-Rad, Laboratories Inc., Hercules, CA, USA) to determine the copy numbers of the *cbbL*, *cbbM* and 16S rRNA genes in DNA gradient fractions. The K2f/V2r primer pair (K2f: 5'-ACC AYC AAG CCS AAG CTS GG-3' and V2r: 5'-GCC TTC SAG CTT GCC SAC CRC-3'), the cbbMF/cbbMR primer pair (cbbMF: 5'-TTC TGG CTG GGB GGH GAY TTY ATY AAR AAY GAC GA-3' and cbbM-R:5'-CCG TGR CCR GCV CGR TGG TAR TG-3') and the

515F/907R primer pair (515F: 5'-GTG CCA GCM GCC GCG G-3' and 907R: 5'-CCG TCA ATT CMT TTR AGT TT-3') were used to amplify the *cbbL*, *cbbM* and 16S rRNA genes, respectively. The reactions were performed in a 20-µL mixture containing 10.0 µL of SYBR Premix Ex Taq (TaKaRa), each primer at 0.5 µM, and 1 µL of DNA template. The amplification efficiencies were 90-100%, which were obtained with $R^2$ values greater than 0.99.

## 2.6 Pyrosequencing of the 16S rRNA gene


The composition of the bacterial communities in different samples was assessed by pyrosequencing of the 16S rRNA genes from triplicate microcosms. The 515F/907R primer pair was used for amplification of the V3-V4 regions of the 16S rRNA gene. The 16S rRNA gene from the [13]C-labeled DNA fraction, with CsCl buoyant densities of 1.738 in the heavy fraction in YM and 1.734 in the heavy fraction in SM, was amplified for pyrosequencing.

Primers were tagged with unique barcodes for each sample. Each sample was amplified in triplicate, and the products were pooled. Negative controls using sterilized water instead of soil DNA extract were included to check for primer or sample DNA contamination. The qualities and concentrations of the purified barcoded PCR products were determined using a NanoDrop spectrophotometer. The bacterial community composition of each sample was assessed by Illumina MiSeq sequencing of the 16S rRNA gene using the MiSeq Reagent Kit v3.



Read merging and quality filtering of the raw sequences were performed using QIIME software with the UPARSE pipeline. The 'identify_chimeric_seqs.py' command was used to identify chimeric sequences according to the UCHIME algorithm, and chimeric sequences were removed with the 'filter_fasta.py' command. Operational taxonomic units (OTUs) were clustered with 97% similarity, and OTU picking and taxonomy assignments were performed with the 'pick_de_novo_otus.py' command for subsequent analysis. The OTUs containing less than 10

reads in the $^{13}C$-labeled DNA fractions were removed. The raw amplicon sequence data of the 16S rRNA genes have been deposited in the GenBank sequence read archive under accession number SRP155504.

### 2.7 Clone library construction of the *cbbL* and *cbbM* genes

Clone libraries of the *cbbL* and *cbbM* genes were also constructed from the $^{13}C$-labeled DNA fractions with CsCl

buoyant densities of 1.738 in the heavy fraction in YM and 1.734 in the heavy fraction in SM. The K2f/V2r and cbbMF/cbbMR primer pairs were used to amplify the *cbbL* and *cbbM* genes, respectively. Triplicate amplicons were pooled, ligated into the pGEM-T vector (Promega, Fitchburg, WI, USA), and transformed into competent DH5α cells. One hundred and eighty-eight *cbbL* gene sequences and one hundred and eighty-three *cbbM* gene sequences were obtained by Sanger sequencing of randomly selected positive clones. OTU clustering with 97% similarity was also

performed with mothur. Representative OTU sequences of the *cbbL* and *cbbM* genes obtained from clone library sequencing have been deposited in GenBank under accession numbers MH699091 to MH699105.

### 2.8 Data analysis

Differences in the overall bacterial community composition among the given samples were visualized by nonmetric

multidimensional scaling (NMDS) using Bray-Curtis distance matrices. The translated *cbbL* and *cbbM* sequences from the heavy fractions were used to construct a phylogenetic tree by the neighbor-joining method using the MEGA package, version 7.0.

### 3. Results

### 3.1 Pyrite oxidation and carbon sequestration


For the two mine tailing control groups, YM_ck and SM_ck, the pH values (3.25±0.09 in YM_ck and 2.98±0.04 in SM_ck), sulfate ($SO_4^{2-}$) contents (13.15±2.58 mg/g in YM_ck and 8.95±2.19 mg/g in SM_ck), and total organic carbon contents (16.75±0.09 mg/g in YM_ck and 18.55±0.12 mg/g in SM_ck) exhibited no significant changes after 14 days of incubation (Fig. 1). The addition of pyrite decreased pH values by 0.48±0.16 and 0.41±0.07 in YM and SM,

respectively. Pyrite addition also increased the $SO_4^{2-}$ content by 252.96% and 262.35%, $Fe^{2+}$ content by 329.47% and 240.38%, and $Fe^{3+}$ content by 137.47% and 140.37% in YM and SM, respectively. Together these data indicate the




occurrence of pyrite oxidization and acidification in mine tailings with the addition of pyrite. Additionally, the TOC content increased by $0.20\pm0.11$ mg/g in YM and $0.28\pm0.14$ mg/g in SM. Furthermore, the $^{13}C$ atom % values in YM_FeS$_2$ and SM_FeS$_2$ were higher than those in the control groups YM_ck and SM_ck, which exhibited $^{13}C$ atom % values of $1.76\pm0.06$ and $1.36\pm0.01$. This result indicated that the fixation of $^{13}C$-$CO_2$ occurred in these mine tailings with the addition of pyrite.

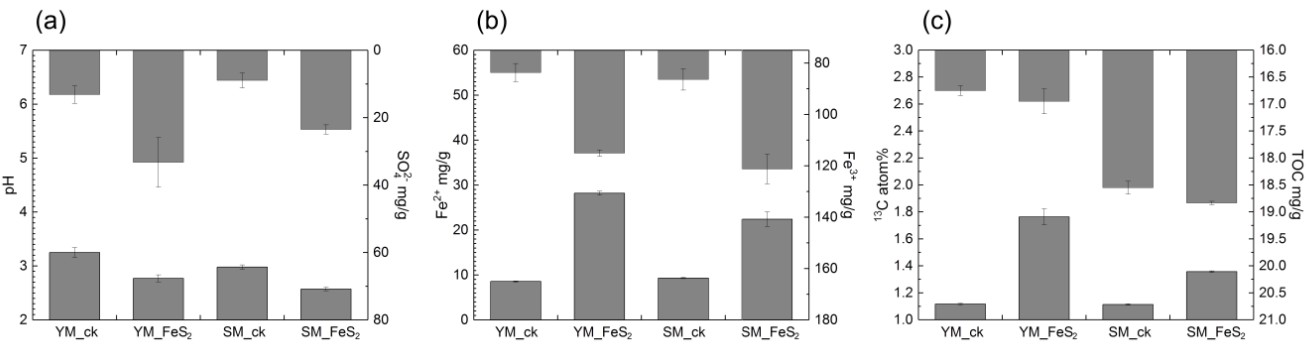

**Fig. 1** pH values, $SO_4^{2-}$ (a), $Fe^{2+}$, $Fe^{3+}$ (b), $^{13}C$ atom % and TOC (c) contents in mine tailings. The error bars indicate the standard errors of three subsamples for each tailing sample. YM_ck, Yangshanchong mine tailings without any treatment; SM_ck, Shuimuchong mine tailings without any treatment; YM_FeS$_2$, Yangshanchong mine tailings treated with FeS$_2$; SM_FeS$_2$, Shuimuchong mine tailings treated with FeS$_2$.

### 3.2 Bacterial communities present during the process of pyrite oxidation

A total of 220877 usable sequences (mean 24541, minimum 9362, maximum 28400) were obtained from total genomic DNA. The ordering of samples by NMDS according to their OTU composition and Bray-Curtis similarity measures (Fig. 2) demonstrated separation of the bacterial community structure in both the YM and SM samples.

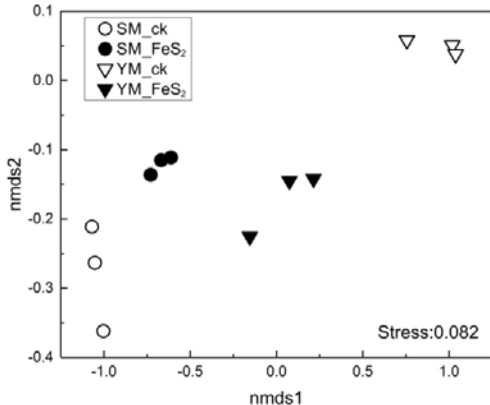

**Fig. 2** Nonmetric multidimensional scaling (NMDS) of the overall bacterial community composition by Bray-Curtis distance matrices in mine tailings.





In this study, a total of 8 bacterial phyla and 4 proteobacterial classes were frequently identified in the two mine

tailings, including Proteobacteria (mainly composed of classes Alphaproteobacteria, Betaproteobacteria,

Gammaproteobacteria and Deltaproteobacteria), Firmicutes, Acidobacteria, Actinobacteria, WPS-2, Planctomycetes,

AD3 and Nitrospirae (Fig. 3). Pyrite addition significantly increased the relative abundances of AD3, Nitrospirae and

unclassified Proteobacteria in YM by 0.75% ($P$=0.008), 0.59% ($P$=0.019) and 6.33% ($P$<0.001), respectively, as well

as significantly increased the relative abundances of Firmicutes, Planctomycetes, unclassified Proteobacteria,

Alphaproteobacteria, Betaproteobacteria, and Deltaproteobacteria by 15.69% ($P$<0.001), 0.97% ($P$<0.001), 5.88%

($P$=0.002), 4.35% ($P$=0.001), 8.61% ($P$<0.001) and 0.21% ($P$=0.003), respectively, and decreased the percentages of

AD3, Acidobacteria, Actinobacteria and Gammaproteobacteria in SM by 0.97% ($P$=0.002), 7.43% ($P$=0.002), 1.35%

($P$=0.016) and 4.85% ($P$=0.002), respectively.

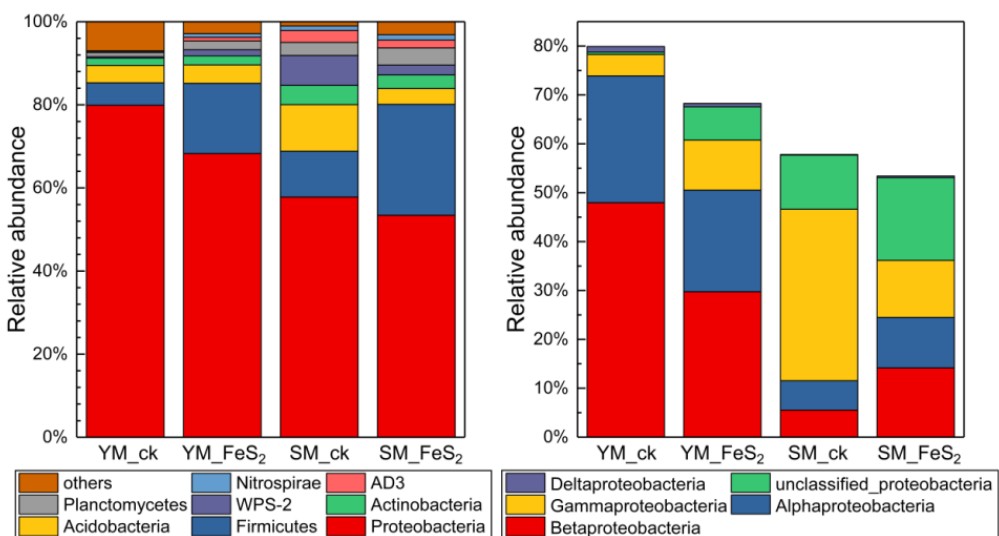


**Fig. 3** Relative abundances (percentages) of the main identified bacterial taxonomic groups, i.e., the phyla
Proteobacteria, Firmicutes, Acidobacteria, Actinobacteria, WPS-2, Planctomycetes, AD3 and Nitrospirae; and the
classes Alphaproteobacteria, Betaproteobacteria, Deltaproteobacteria and Gammaproteobacteria (within the phylum
Proteobacteria). For each tailing sample, the relative abundances of the sequences assigned to a given taxonomic unit
were calculated for each of three subsamples, and the average value was then used to represent the relative abundance
of each tailing sample.





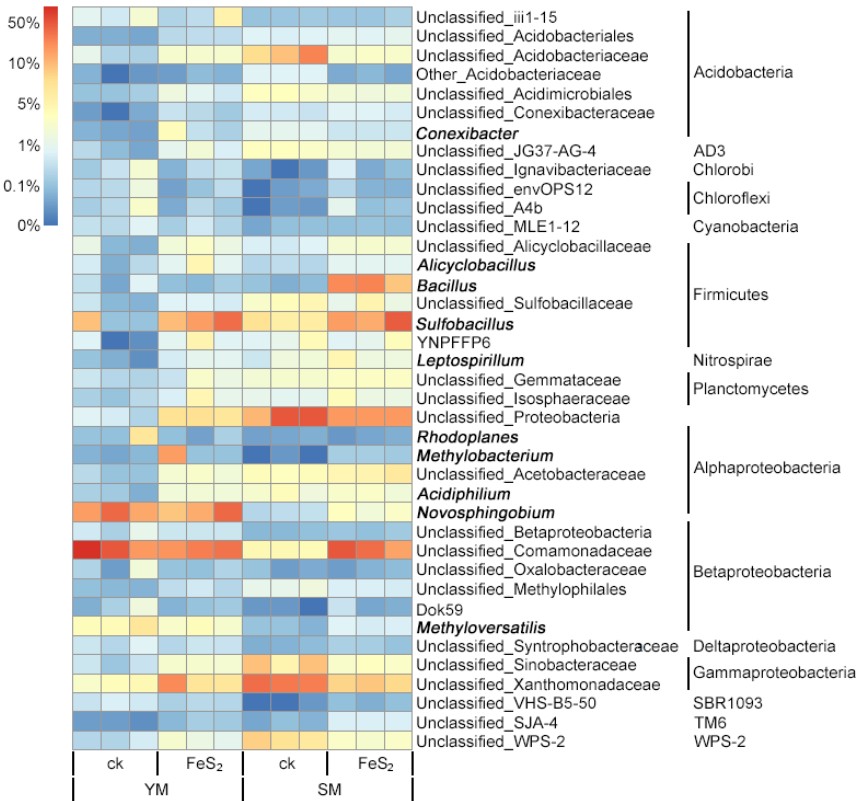

**Fig. 4** Heatmap of the top genera with relative abundances above 0.2% in mine tailings.

We constructed a heatmap diagram (Fig. 4) that shows the top 39 dominant genera with relative abundances above 0.20% from the mine tailings. However, only a small number of genera were assigned to known taxa (accounting for 27.50% of total bacterial communities), including *Alicyclobacillus*, *Bacillus*, *Sulfobacillus*, YNPFFP6, *Leptospirillum*, *Rhodoplanes*, *Methylobacterium*, *Acidiphilium*, *Novosphingobium*, Dok59, and *Methyloversatilis* (Fig. 4). The *Sulfobacillus* (8.04%) and *Novosphingobium* (8.60%) genera accounted for 16.64% of the total bacterial communities and were the dominant taxa in the mine tailings. In addition, ferrous and sulfur-oxidizing bacteria, *Leptospirillum* and *Acidithiobacillus*, accounted for merely 0.80% and 0.02% of the total bacterial communities, respectively. In addition, pyrite addition significantly increased the relative abundances of the genera *Alicyclobacillus*, *Leptospirillum*, *Sulfobacillus* and *Acidiphilium* in YM by 0.67% (*P*=0.027), 0.74% (*P*=0.002), 8.86% (*P*=0.043) and 1.57% (*P*<0.001), respectively, as well as significantly increased the relative abundances of *Alicyclobacillus*, *Bacillus*, *Sulfobacillus*, *Methylobacterium*, *Novosphingobium* and *Methyloversatilis* in SM by 0.57% (*P*<0.001), 9.94% (*P*<0.001), 5.99% (*P*<0.001), 0.15% (*P*<0.001), 2.06% (*P*=0.004) and 0.56% (*P*<0.001), respectively.



## 3.3 DNA-SIP for autotroph identification in mine tailings

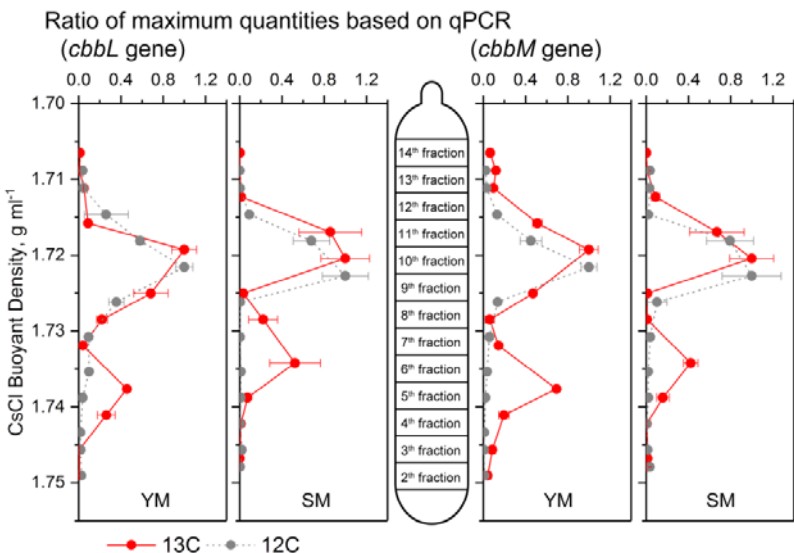

**Fig. 5** Quantitative distribution of *cbbL* and *cbbM* gene fragments across the entire buoyant density gradients of the
DNA fractions from microcosms treated with $FeS_2$ and incubated with $^{12}C$-$CO_2$ or $^{13}C$-$CO_2$. The normalized data
consist of the ratio of the gene copy number for each DNA gradient to the maximum quantity for each treatment. The
error bars represent the standard errors of triplicate microcosms, and each contains three technical replicates. YM,
Yangshanchong mine tailings; SM, Shuimuchong mine tailings.

For the quantitative analysis of *cbb* gene abundance, the buoyant densities of the DNA in isopycnic centrifugation
gradients were used to assess the labeling efficiencies of *cbb* gene-carrying carbon fixers in the DNA-SIP microcosms
(Fig. 5). A shift toward heavy fractions was observed for the *cbbL* and *cbbM* gene abundances with $^{13}C$-$CO_2$ treatment
but not $^{12}C$-$CO_2$ control treatment (Fig. 5). *cbbL* and *cbbM* gene levels under $^{13}C$-$CO_2$ treatment peaked in the heavy
fraction, with buoyant densities of 1.738 g·mL$^{-1}$ in YM and 1.734 g·mL$^{-1}$ in SM. By contrast, the highest copy numbers
of the *cbbL* and *cbbM* genes under $^{12}C$-$CO_2$ treatment appeared in the light fraction, with a buoyant density of
1.722-1.723 g·mL$^{-1}$. These results demonstrated that a considerable amount of $^{13}C$-$CO_2$ was assimilated by carbon
fixers in $^{13}C$-$CO_2$-labeled mine tailing samples, leading to a significant shift of *cbb* gene-carrying genomic DNA into
the heavy fraction.

The *cbbL* and *cbbM* gene sequences from *cbb* clone libraries in the $^{13}C$-DNA heavy fraction treated with $^{13}C$-$CO_2$
were used for phylogenetic analysis (Fig. 6). For the RubisCO form I and II-coding genes, the *cbbL* gene copy
numbers were 16.50-fold and 8.20-fold higher than the *cbbM* gene copy number in the heavy fractions of YM and SM,
respectively. In addition, a vast majority of the *cbb* genes may be associated with unknown groups, with the exception
of cbbL-OTU1, cbbL-OTU6, cbbL-OTU9, cbbM-OTU5 and cbbM-OTU6. These OTUs were associated with
*Sulfobacillus*, *Acidithiobacillu*s and *Azospirillum* in the *cbbL* gene and *Acidithiobacillu*s and *Thiobacillus* in the *cbbM*



gene according to phylogenetic analysis based on amino acid sequences. The *Sulfobacillus*-like *cbbL* gene sequence, cbbL-OTU1, accounted for 30.11% and 34.74% of the total of *cbbL* gene sequences in the heavy fractions of YM and SM, respectively. The *Acidithiobacillus*-like *cbb* genes, including *cbbL* and *cbbM,* accounted for merely 2.15% and 4.21% of the total *cbbL* gene sequences and 3.30% and 4.35% of the total *cbbM* gene sequences in the heavy fractions of YM and SM, respectively. In addition, the *Sulfobacillus* genus had the highest relative abundance compared to any

other genus based on 16S rRNA analysis (Table S1), accounting for 17.18% and 18.24% of the heavy fractions of YM and SM, respectively. The *Leptospirillum* genus accounted for 1.32% and 1.58% of the heavy fractions of YM and SM, respectively, and *Acidithiobacillus* accounted for 0.11% and 0.06% of the heavy fractions of YM and SM, respectively.

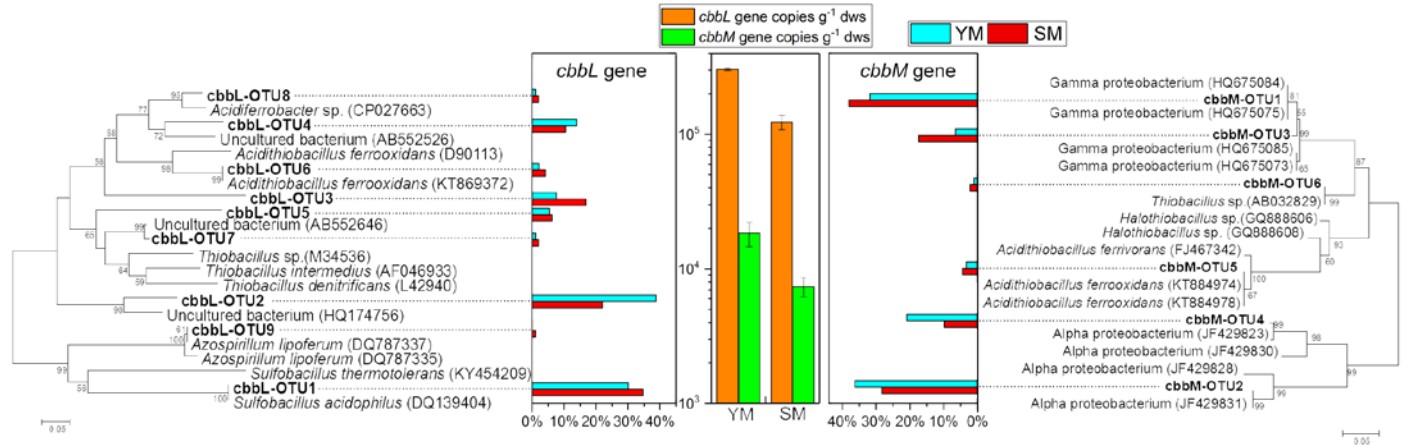

**Fig. 6** Phylogenetic tree of the translated *cbbL* and *cbbM* sequences in the heavy fractions from YM and SM treated
with FeS$_2$. Relative frequencies (%) are marked in the bar graph. Bootstrap values of >50% are indicated at branch points. The *cbbL* and *cbbM* gene copy numbers in the heavy fractions from FeS$_2$-treated YM and SM are shown in the middle of the figure.

## 4. Discussion

Acidic polymetallic mine tailings have strong potential for pyrite oxidation. In this study, a large amount of sulfuric acid was generated (increases of approximately 19.95 mg/g and 14.64 mg/g in YM and SM, respectively), and a persistent decline in pH was observed (pH decreased by 0.44 and 0.35 in YM and SM, respectively) in only 14 days. Most strikingly, pyrite oxidation or acidification of polymetallic mine tailings did not lead to the death of all microbes. The $^{13}$C content and total organic carbon content in mine tailings were found to increase slightly; therefore, it is

obvious that the activities of autotrophic microbes in mine tailings immobilized atmospheric CO$_2$.

### 4.1 Carbon sequestration in mine tailings

Previous studies showed that mine tailings provide an excellent substrate for carbon sequestration through the



formation of carbonate, due to the large surface area of the material grains (McCutcheon et al., 2016). Compared to
soils and natural bedrock, mine tailings may possess higher carbonate precipitation rates (Wilson et al., 2009). In
addition, microbial photosynthesis accelerates carbonate mineral precipitation and further induces mineralization
(McCutcheon et al., 2014;McCutcheon et al., 2016). However, the carbon sequestration process when the activities of
autotrophic organisms are present has not been directly reported. To the best of our present knowledge, this report is
the first to elucidate carbon sequestration by autotrophic groups in mine tailings based on isotope tracers and DNA-SIP.
In terrestrial ecosystems, carbon sequestration occurs mainly via plant photosynthesis and litter decomposition by
microbes (Antonelli et al., 2018). These activities are beneficial to advance vegetative growth, improve the
environment and mitigate global warming (Ghosh et al., 2018;Piccolo et al., 2018). However, no vegetation grows in
the organic carbon-limited habitat of mine tailings. Such extreme environments are considered similar to Earth's early
conditions (Paerl et al., 2000;Liu et al., 2014). Extremophiles dominate autotrophic groups. These organisms fix
atmospheric $CO_2$ into substrate, allowing for increased organic carbon in the soil and providing a carbon source for
heterotrophic microorganisms in this microenvironment. A large number of carbonate minerals in mine tailings can be
transformed by microbes, which accelerates the conversion of inorganic carbon to organic carbon.

### 4.2 Autotrophic bacteria in mine tailings

Previous studies considered that the growth of microorganisms in bare mine tailings is usually limited by the
availability of organic carbon (Schimel and Weintraub, 2003). In this study, the bacterial composition in different mine
tailings varied greatly. Once pyrite was added and the acidification of mine tailings accelerated, some specific taxa,
including the Firmicutes phylum, the *Sulfobacillus*, *Leptospirillum* and *Acidiphilium* genera and many unclassified
taxa, increased in both of the tested mine tailings, showing the high consistency of bacterial community composition in
different mine tailings. It is possible that the availability of organic carbon can stimulate particular carbon fixers (Deng
et al., 2016;Antonelli et al., 2018), and the main autotrophic bacteria found in these two mine tailings may be derived
from the same species. However, a majority of the OTUs observed could not be assigned to known groups.

During the acidification process in mine tailings, some acidophilic and autotrophic microorganisms have very high
activity levels. For example, the *Sulfobacillus* and *Leptospirillum* genera, both of which are vital ferrous and sulfur
oxidizers, increased significantly during pyrite oxidization. Zhang et al. (2016) found genes for the CBB pathway and
rTCA, but no other $CO_2$ fixation pathways, in a copper bioleaching microbial community. For the CBB pathway in this
study, the *Sulfobacillus*-like *cbbL* gene was the primary carbon fixing-associated gene. Previous studies have
confirmed the presence of *Sulfobacillus* in mine tailings (Coral et al., 2018;Yu et al., 2018); *Sulfobacillus* has the
ability to oxidize or reduce Fe(III) and oxidize sulfur (Dold et al., 2005). This ability is important, as this genus likely
leads to a high mineral dissolution rate by adhering to mineral surfaces and further enhancing sulfide mineral oxidation





(Li et al., 2016;Becker et al., 2011). None of the *cbb* genes identified were highly homologous to genes in *Leptospirillum*, but this may be due to primer specificity. Nevertheless, Sabrina Marín et al. (2017) found that the rTCA carbon fixation pathway genes were mainly found in by *Leptospirillum* spp. RubisCO is the most prominent enzyme, and the gene coding for the large subunit of RubisCO serves as a marker for the analysis of autotrophic

organisms, including bacteria, using the CBB cycle (Berg, 2011). In addition, the *Sulfobacillus*-like *cbbL* gene dominated the $^{13}$C-lableled DNA of the carbon-fixing taxa. Furthermore, the higher relative abundance of *Sulfobacillus* than *Leptospirillum,* according to 16S rRNA analysis, demonstrates the contribution of the *Sulfobacillus*-like *cbbL* gene to carbon sequestration. Even so, while the number (or relative abundance) of autotrophs demonstrated their ability to sequester carbon, it did not reflect their ability to perform or their importance in ferrous and sulfur oxidation.

For example, the reduced percentage of the genus *Acidithiobacillus* in the two mine tailings did not reflect the contribution of this genus to the oxidation of iron and sulfur. Falagán et al. (2017) highlighted the importance of thermo-tolerant acidophiles, such as *Acidithiobacillus* and the genus *Sulfobacillus,* in extracting and recovering metals from mine tailings. Furthermore, it has been known for many years that *Acidithiobacillus* can obtain energy by catalyzing the oxidation reaction of $Fe^{2+}$ to $Fe^{3+}$ from sulfites (Dold, 2014), and this may significantly speed up the rate

of ferrous oxidization. On the other hand, *Acidithiobacillus* can use the monodentate low-molecular-weight carboxylic acids (LMWCA) as electron donors and ferric iron as an electron sink, resulting in ferrous oxidization (Dold et al., 2005;Dold, 2014). The decreased level of total carbon, including LMWCA, may also limit the activity of this bacterium.

Archaea may also be an important participant in the acidification of mine tailings. Chen et al. (2013) and Liu et al.

(2014) found that the relative abundances of Euryarchaeota belonging to the archaea significantly increased with decreasing pH, which indicates that this taxon is an indicator of metal contamination (Hur et al., 2011). Despite this, clone libraries of the *cbbL* and *cbbM* genes in the $^{13}$C-labeled heavy fraction did not show archaeal sequences for Calvin cycle genes. However, the archaea may have higher activities in RubisCO-mediated carbon metabolic pathways (Kono et al., 2017), which will require further study.

In conclusion, this study was the first to elucidate carbon sequestration by autotrophic groups in mine tailings based on isotope tracers and DNA-SIP. Our results demonstrated higher $^{13}$C atom % values with the addition of pyrite than in control groups after a 14-day incubation, as well as a significant increase in the total organic carbon content. The *Sulfobacillus* genus was dominant in the pyrite-treated bacterial communities and was also the primary carbon fixer with the RubisCO form I-encoding gene *cbbL*. Finally, the *cbbL* gene may play a vital role in carbon sequestration in

the sulfide mineral oxidation of mine tailings.

**Acknowledgments** This study was funded by the National Natural Science Foundation of China (31800456), the



National Key Basic Research Program of China (2015CB150506), the Strategic Priority Research Program of the CAS (XDB15040000), the Natural Science Foundation of the Anhui Provincial Education Department (KJ2018A0032), and the PhD Research Startup Foundation of Anhui University (J01003269).

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
