# Peer review of "Pyrite oxidization accelerates bacterial carbon sequestration in copper mine tailings"

_Biogeosciences, 2018_

## Referee Comment (RC1) · Anonymous Referee #1 · 11 Sep 2018

1. the section of 2.1 should include more details about the soil sampling. 2. The experiment treatments was also not clear. From section 2.2, we can't judge how many treatments were set. Please specify. Moreover, what's samples were conducted qPCR, sequencing and cloning were also confused. 3. For section 4.1, authors only described how Carbon sequestration in mine tailings but without any discussion Combining with your own data.

---

## Referee Comment (RC2) · Anonymous Referee #2 · 18 Sep 2018

The manuscript explored the autotrophic microbes and the FeS2 facilitation role in acidic mine tailings using stable isotope and molecular methods. The results showed that FeS2 facilitated CO2-fixing by microbes and increased the abundances of relevant autotrophs. The study is very interesting, which could provide new insights into the autotrophic roles in extreme environments. However, the article writing is awful in logic, result description and interpretation. Here are my concerns: 1 The introduction did not show some key points relevant to the research, such as possible CO2-fixing pathways and autotrophs in acidic sulfur-enriched environments. The introduction was not well structured and really needs rewrite. 2 The method section failed to describe key details: 1) weather the samples were washed by acid prior to measuring isotope compositions?

[Figure]

2) no descriptions on chemical analysis in samples, i.e. solutes for Fe2+,Fe3+, SO42-
.... 3) no citations for the primer sets, which were apparrently designed in the study
4) no informations on PCR reactions 5) how did the authors determine the PCR ef-
ficiency? 6) how did the authors qualify gene abundance? standard curves? 7) no
statistic software informations 8) how many replicates were for each treatment? 3 Fig1
symbols are very confusing, and no descriptions on the above and bottom columns.
4 No specific legends or descriptions on the two inserts in Fig. 3, and the color dif-
ferences are not clear. 5 L252-261, 284-291, there are many super long sentences.
A sentence usually contains maximum 22 words. 6 The CO2-fixing capacity by au-
totrophs should be calculated. 7 L306-307, the statement is problematic: 12CO2 is
a control relative to 13CO2, so the shift to heavy fractions should not be observed in
12CO2. 8 L307-311, the statements are not correct: for the peak in 13CO2 occurred
in the density of 1.72 rather than 1.73 in both 12CO2 and 13CO2. 9 L311-314, the
statements should go to discussion section. 10 Fig. 6, Cultured genus most related to
OTU1, 2, 3 and 4 should be given for identifying purpose. 11 cbb is not a correct gene
name, it should be cbbL or cbbM. 12 Is Fig. 5 for FeS2 treatments or raw mine tailings?
13 L351-371, the paragraph should go to introduction section. 14 The discussion is far
from the results, i.e. discuss why and how FeS2 facilitates microbial CO2-fixing and
changes the whole bacterial community.

---

## Author Comment (AC2) · 20 Oct 2018

1. The manuscript explored the autotrophic microbes and the FeS2 facilitation role in acidic mine tailings using stable isotope and molecular methods. The results showed that FeS2 facilitated CO2-fixing by microbes and increased the abundances of relevant autotrophs. The study is very interesting, which could provide new insights into the autotrophic roles in extreme environments. However, the article writing is awful in logic, result description and interpretation. Reply: We thank the reviewer for pointing out the importance of this study and giving me a number of useful advices.

2. Comments from Referees: The introduction did not show some key points relevant to the research, such as possible $CO_2$-fixing pathways and autotrophs in acidic sulfur-enriched environments. The introduction was not well structured and really needs rewrite. Response: Thanks for this comment. I have rewritten the introduction details according to this comment. Changes in manuscript: (Lines 29-61) "Terrestrial ecosystems have great potential as carbon sinks to stabilize $CO_2$ and regulate climate change (White et al., 2000), and atmospheric $CO_2$ can be fixed into plants by photosynthesis and assimilated into soils as soil organic carbon (SOC) by decomposition and microbial activity (Deng et al., 2016;Antonelli et al., 2018). Currently, chemolithoautotrophic organisms fix atmospheric $CO_2$ by six pathways, including the widely distributed Calvin-Benson-Bassham (CBB) cycle, the reductive tricarboxylic acid (rTCA) cycle, the reductive acetyl-CoA pathway and the recently discovered 3-hydroxypropionate/4-hydroxybutyrate (HP/HB) cycles (Berg, 2011;Alfreider et al., 2017). The ribulose-1,5-bisphosphate carboxylase/oxygenase (RuBisCO) enzyme is the most prominent enzyme on earth (Raven, 2013) and is important in the CBB cycle. The CBB cycle is the most prevalent means of $CO_2$ fixation by autotrophs (Tabita, 1999;Berg, 2011). In addition, the genes encoding the large subunit of RuBisCO serve as autotroph markers (Alfreider et al., 2017;Berg, 2011). The cbbL and cbbM genes encode RuBisCO form I and form II, respectively, with 25 to 30% amino acid sequence identity (Tabita et al., 2008). Polymetallic mine tailings have considerable potential to stabilize levels of atmospheric $CO_2$ (Harrison et al., 2013) through the carbonation of noncarbonate minerals, including the dissolution of silicates, hydroxides and oxides and the precipitation of carbonate minerals (McCutcheon et al., 2016;Meyer et al., 2014;McCutcheon et al., 2014). However, sequestered carbon is also present in the form of inorganic carbon. Compared with soil ecosystems, polymetallic mine tailings exhibit specific features, including a lack of organic matter, nutrients and nutrient-holding capacity (Lottermoser, 2010;Young et al., 2015), that restrict plant growth, and plant productivity is generally difficult to restore in mining wastelands (Hu et al., 2018;Li et al., 2017). Therefore, microbes may be the main source of organic carbon in mine tailings. The limited amount of organic matter in mine tailings also inhibits the activities of heterotrophic microorganisms, and therefore the microorganisms in these environments are lithotroph-dominant (Li et al., 2015). Polymetallic mine tailings contain sulfide minerals (e.g., pyrite), and the oxidation of these sulfide minerals leads to a decrease in pH, also known as mine tailing acidification. Previous studies have noted that polymetallic mine tailings have lithotroph-dominant microbial compositions (Li et al., 2015) due to the limited amount of organic matter. Consequently, acidophilic, chemoautotrophic bacteria, including Acidithiobacillus, Leptospirillum and Sulfobacillus (Chen et al., 2013;Liu et al., 2014), can promptly participate in ferrous and sulfur oxidation in mine tailings, and these autotrophic taxa play leading roles in carbon cycling and energy flow during the mine tailing acidification process. However, the relationship between the oxidation of sulfide minerals and carbon sequestration by these acidophilic, chemoautotrophic bacteria is still unknown. In the present study, we conducted a microcosm experiment using mine tailings collected from two mines to determine the effects of sulfide mineral (pyrite) oxidation on carbon sequestration in mine tailings through the addition of pyrite. In addition, the main carbon fixers were analyzed through DNA-based stable isotope probing (DNA-SIP) and cbbL and cbbM gene analysis. Our objectives were to investigate whether sulfide mineral oxidation can stimulate carbon sequestration in mine tailings and to identify the key carbon sequestration groups in mine tailings during the acidification process."

3. Comments from Referees: The method section failed to describe key details: 1) weather the samples were washed by acid prior to measuring isotope compositions? 2) no descriptions on chemical analysis in samples, i.e. solutes for Fe2+,Fe3+, SO42-; 3) no citations for the primer sets, which were apparently designed in the study; 4) no informations on PCR reactions 5) how did the authors determine the PCR efficiency? 6) how did the authors qualify gene abundance? standard curves? 7) no statistic software informations; 8) how many replicates were for each treatment? Response: Thanks for this comment. I have rewritten and added more information according to this comment. Changes in manuscript: (1) (Lines 85-86) "Carbon isotope composition was analyzed by a Delta V Advantage Mass Spectrometer (Thermo Fisher Scientific, Inc.,

USA) coupled with an elemental analyzer (Flash2000; HT Instruments, Inc., USA) in continuous flow mode." (Lines 89-91) "The carbon isotope composition and TOC content were analyzed after performing a soil acidification pretreatment method to remove inorganic carbon as described previously (Wang et al., 2015)" (2) (Lines 92-97) "The $Fe^{2+}$ and $Fe^{3+}$ in the soils were extracted by HCl. The $Fe^{2+}$ in the extract was measured by a spectrophotometric method after mixing with phenanthroline and trisodium citrate, and the $Fe^{3+}$ in the extract was reduced to $Fe^{2+}$ by hydroxylammonium chloride and measured by a spectrophotometric method (Heron et al., 1994). The total sulfate ion content was determined via ion chromatography after extraction by sodium hydroxide, as described previously (Yin and Catalan, 2003)." (3) (Lines 109-114) "The K2f/V2r primer pair (K2f: 5'-ACC AYC AAG CCS AAG CTS GG-3' and V2r: 5'-GCC TTC SAG CTT GCC SAC CRC-3') (Nanba et al., 2004), the cbbMF/cbbMR primer pair (cbbMF: 5'-TTC TGG CTG GGB GGH GAY TTY ATY AAR AAY GAC GA-3' and cbbM-R: 5'-CCG TGR CCR GCV CGR TGG TAR TG-3') (Campbell and Cary, 2004) and the 515F/907R primer pair (515F: 5'-GTG CCA GCM GCC GCG G-3' and 907R: 5'-CCG TCA ATT CMT TTR AGT TT-3') (Zhou et al., 2011) were used to amplify the cbbL, cbbM and 16S rRNA genes, respectively." (4) (Lines 116-117) "qPCR analysis of the cbbL, cbbM and 16S rRNA genes was performed under the following conditions: 40 cycles of 30 s at 95°C, 30 s at 55°C (cbbL and 16S rRNA genes) or 57°C (cbbM gene), and 45 s at 72°C." (5) (Lines 117-120) "Standard curves were obtained using 10-fold serial dilutions of linearized recombinant plasmids containing the cbbL, cbbM and 16S rRNA genes with known copy numbers. The amplification efficiencies were 90-100%, which were obtained with $R^2$ values greater than 0.99." (6) (Lines 117-119) "Standard curves were obtained using 10-fold serial dilutions of linearized recombinant plasmids containing the cbbL, cbbM and 16S rRNA genes with known copy numbers." (7) (Lines 151-156) "Bray-Curtis distance matrices for the overall bacterial community composition among the given samples were calculated in R v.3.3.2 using the 'vegdist' function of the vegan package and visualized by nonmetric multidimensional scaling (NMDS) in Origin 8. A heatmap of dominant genera with relative abundances above

0.2% was applied for plotting in the R environment with the pheatmap package. The translated cbbL and cbbM sequences from the heavy fractions were used to construct a phylogenetic tree by the neighbor-joining method using the MEGA package, version 7.0." (8) (Lines 175-182) "There were a total of four treatments in the microcosms of the two mine tailings. For each mine tailing, fresh mine tailings (equivalent to 10.0 g d.w.s.) were mixed with a total of 2 g of sterile pulverized $FeS_2$ at approximately 60% maximum water-holding capacity as the $FeS_2$ treatment, followed by incubation at 25°C in the dark for 14 days. Yangshanchong mine tailing samples (YM) cultured with $FeS_2$ are abbreviated as YM_FeS2, and Shuimuchong mine tailing samples (SM) cultured with $FeS_2$ are abbreviated as SM_FeS2. In addition, fresh mine tailings at approximately 60% maximum water-holding capacity without any additive were used as the control groups and abbreviated as YM_ck and SM_ck. For each treatment, the microcosms were incubated with 10% $^{13}C$-$CO_2$ or $^{12}C$-$CO_2$, and both treatments were constructed in triplicate for DNA-SIP analysis."

4. Comments from Referees: Fig1 symbols are very confusing, and no descriptions on the above and bottom columns. Response: Thanks for this comment. I have redrawn the figure and have rewritten the figure legend. Changes in manuscript:

Fig. 1 pH values (a), $^{13}C$ atom % (b), TOC (c) contents, $SO_4^{2-}$ (d), $Fe^{2+}$ (e) and $Fe^{3+}$ (f) in mine tailings. The error bars indicate the standard errors of three subsamples for each tailing sample. To determine $^{13}C$ atom % (b), all analyzed samples were treated with $^{13}C$-$CO_2$ in microcosms. YM_ck, control group of Yangshanchong mine tailings; SM_ck, control group of Shuimuchong mine tailings; YM_FeS2, Yangshanchong mine tailings treated with $FeS_2$; SM_FeS2, Shuimuchong mine tailings treated with $FeS_2$.

5. Comments from Referees: No specific legends or descriptions on the two inserts in Fig. 3, and the color differences are not clear Response: Thanks for this comment. I have redrawn the figure and have rewritten the figure legend. Changes in manuscript:

Fig. 3 Relative abundances (percentages) of the main identified bacterial taxonomic

groups, i.e., the phyla Proteobacteria, Firmicutes, Acidobacteria, Actinobacteria, WPS-2, Planctomycetes, AD3 and Nitrospirae (a); and the classes Alphaproteobacteria, Betaproteobacteria, Deltaproteobacteria and Gammaproteobacteria (within the phylum Proteobacteria) (b). For each tailing sample, the relative abundances of the sequences assigned to a given taxonomic unit were calculated for each of three subsamples, and the average value was then used to represent the relative abundance of each tailing sample.

6. Comments from Referees: L252-261, 284-291, there are many super long sentences. A sentence usually contains maximum 22 words. Response: Thanks for this comment. I have rewritten these sentences. Changes in manuscript: (1) (Lines 190-197) "In the Yangshanchong mine tailings, pyrite addition significantly increased the relative abundances of AD3, Nitrospirae and unclassified Proteobacteria by 0.75% (P=0.008), 0.59% (P=0.019) and 6.33% (P<0.001), respectively. In parallel, in Shuimuchong mine tailings, FeS2 addition significantly increased the relative abundances of Firmicutes, Planctomycetes, unclassified Proteobacteria, Alphaproteobacteria, Betaproteobacteria, and Deltaproteobacteria by 15.69% (P<0.001), 0.97% (P<0.001), 5.88% (P=0.002), 4.35% (P=0.001), 8.61% (P<0.001) and 0.21% (P=0.003), respectively. However, in SM, the percentages of AD3, Acidobacteria, Actinobacteria and Gammaproteobacteria in SM by 0.97% (P=0.002), 7.43% (P=0.002), 1.35% (P=0.016) and 4.85% (P=0.002) were decreased under pyrite addition." (2) (Lines 214-220) "In the Yangshanchong mine tailings, pyrite addition significantly increased the relative abundances of the genera Alicyclobacillus, Leptospirillum, Sulfobacillus and Acidiphilium by 0.67% (P=0.027), 0.74% (P=0.002), 8.86% (P=0.043) and 1.57% (P<0.001), respectively. In the Shuimuchong mine tailings, FeS2 addition significantly increased the relative abundances of Alicyclobacillus, Bacillus, Sulfobacillus, Methylobacterium, Novosphingobium and Methyloversatilis by 0.57% (P<0.001), 9.94% (P<0.001), 5.99% (P<0.001), 0.15% (P<0.001), 2.06% (P=0.004) and 0.56% (P<0.001), respectively."

7. Comments from Referees: The CO2-fixing capacity by autotrophs should be calculated. Response: Thanks for this comment. I have calculated the $CO_2$-fixing capacity by autotrophs in the result. Changes in manuscript: (Lines 170-171) "The $CO_2$-fixing capacities of autotrophs under $FeS_2$ addition were 9.50±0.91 mg/kgÂůd in YM and 3.69±0.11 mg/kgÂůd in SM."

8. Comments from Referees: L306-307, the statement is problematic: $12CO_2$ is a control relative to $13CO_2$, so the shift to heavy fractions should not be observed in $12CO_2$. L307-311, the statements are not correct: for the peak in $13CO_2$ occurred in the density of 1.72 rather than 1.73 in both $12CO_2$ and $13CO_2$. Response: Thanks for this comment. I have written the sentences. Changes in manuscript: (Lines 232-235) "cbbL and cbbM gene levels under 13C-$CO_2$ treatment peaked at a density of 1.72 in both the 12C-$CO_2$ and 13C-$CO_2$ treatments. In addition, a shift toward heavy fractions was observed for the cbbL and cbbM gene abundances in the 13C-$CO_2$ treatment (Fig. 5), with buoyant densities of 1.738 gÂůmL-1 in YM_FeS2 and 1.734 gÂůmL-1 in SM_FeS2."

9. Comments from Referees: L311-314, the statements should go to discussion section Response: Thanks for this comment. I have move the works into the discussion section. Changes in manuscript: (Lines 301-306) "In addition, only a few archaea were detected based on 16S rRNA gene sequencing, and the clone libraries of the cbbL and cbbM genes in the 13C-labeled heavy fraction did not show archaeal sequences for Calvin cycle genes. These results indicated that bacterial carbon sequestration is mainly attributable to chemoautotrophic bacteria in pyrite oxidization of mine tailings. However, archaea may have higher activities in RuBisCO-mediated carbon metabolic pathways (Kono et al., 2017), which will require further study."

10. Comments from Referees: Fig. 6, Cultured genus most related to OTU1, 2, 3 and 4 should be given for identifying purpose Response: Thanks for this comment. I have added more information about the cultured genus most related to OTU1, 2, 3 and 4 in the supply materials Changes in manuscript: Fig. 6 Phylogenetic tree of the translated cbbL and cbbM sequences in the heavy fractions from YM and SM treated

with FeS2. Relative frequencies (%) are marked in the bar graph. Bootstrap values of >50% are indicated at branch points. The cbbL and cbbM gene copy numbers in the heavy fractions from FeS2-treated YM and SM are shown in the middle of the figure. The cultured genera most related to OTUs from the cbbL and cbbM clone libraries are shown in Table S1.

Table S1 Cultured genus most related to OTUs from cbbL and cbbM clone libraries

11. Comments from Referees: cbb is not a correct gene name, it should be cbbL or cbbM. Response: Thanks for this comment. I have rewritten the gene name across the manuscript. Changes in manuscript: (Lines 237-238) "The cbbL and cbbM gene sequences from the clone libraries in the 13C-DNA heavy fraction treated with 13C-CO2 were used for phylogenetic analysis (Fig. 6)." (Lines 230-232) "For the quantitative analysis of cbbL and cbbM gene abundances, the buoyant densities of the DNA in isopycnic centrifugation gradients were used to assess the labeling efficiencies of cbbL or cbbM gene-carrying carbon fixers in the DNA-SIP microcosms (Fig. 5)."

12. Comments from Referees: Is Fig. 5 for FeS2 treatments or raw mine tailings? Response: Thanks for this comment. The figure was for FeS2 treatment and I have redrawn the figure. Changes in manuscript:

Fig. 5 Quantitative distribution of cbbL and cbbM gene fragments across the entire buoyant density gradients of the DNA fractions from microcosms treated with FeS2 and incubated with 12C-CO2 or 13C-CO2. The normalized data consist of the ratio of the gene copy number for each DNA gradient to the maximum quantity for each treatment. The error bars represent the standard errors of triplicate microcosms, and each contains three technical replicates.

13. Comments from Referees: L351-371, the paragraph should go to introduction section. The discussion is far from the results, i.e. discuss why and how FeS2 facilitates microbial CO2-fixing and changes the whole bacterial community. Response: Thanks for this comment. I have rewritten the Introduction and Discussion

according to this comment. Changes in manuscript: (1) Introduction: "Terrestrial 
[revised manuscript text omitted]

Please also note the supplement to this comment:
https://www.biogeosciences-discuss.net/bg-2018-370/bg-2018-370-AC2-supplement.pdf
* * *
[Figure]

[Figure]

Fig. 1 pH values (a), $^{13}$C atom % (b), TOC (c) contents, $SO_4^{2-}$ (d), $Fe^{2+}$ (e) and $Fe^{3+}$ (f) in mine tailings. The error bars indicate the standard errors of three subsamples for each tailing sample. To determine $^{13}$C atom % (b), all analyzed samples were treated with $^{13}$C-$CO_2$ in microcosms. YM_ck, control group of Yangshanchong mine tailings; SM_ck, control group of Shuimuchong mine tailings; YM_FeS$_2$, Yangshanchong mine tailings treated with FeS$_2$; SM_FeS$_2$, Shuimuchong mine tailings treated with FeS$_2$.

[Figure]

Fig. 3 Relative abundances (percentages) of the main identified bacterial taxonomic groups, i.e., the phyla Proteobacteria, Firmicutes, Acidobacteria, Actinobacteria, WPS-2, Planctomycetes, AD3 and Nitrospirae (a); and the classes Alphaproteobacteria, Betaproteobacteria, Deltaproteobacteria and Gammaproteobacteria (within the phylum Proteobacteria) (b). For each tailing sample, the relative abundances of the sequences assigned to a given taxonomic unit were calculated for each of three subsamples, and the average value was then used to represent the relative abundance of each tailing sample.

[Figure]

**Fig. 5** Quantitative distribution of *cbbL* and *cbbM* gene fragments across the entire buoyant density gradients of the DNA fractions from microcosms treated with FeS$_2$ and incubated with $^{12}$C-CO$_2$ or $^{13}$C-CO$_2$. The normalized data consist of the ratio of the gene copy number for each DNA gradient to the maximum quantity for each treatment. The error bars represent the standard errors of triplicate microcosms, and each contains three technical replicates.

**Supplement:**

Table S1 Cultured genus most related to OTUs from *cbbL* and *cbbM* clone libraries

| cbbL OTU | Closest affiliation | Max identity | cbbM OTU | Closest affiliation | Max identity |
|---|---|---|---|---|---|
| cbbL-OTU1 | *Sulfobacillus acidophilus* (DQ139404) | 99 | cbbM-OTU1 | Gamma proteobacterium (HQ675075) | 98 |
| cbbL-OTU2 | *Thiohalospira halophila* (GQ888600) | 83 | cbbM-OTU2 | *Magnetospirillum moscoviense* (KF712469) | 88 |
| cbbL-OTU3 | *Chromatium vinosum* (M26396) | 78 | cbbM-OTU3 | Gamma proteobacterium (HQ675084) | 98 |
| cbbL-OTU4 | *Acidithiobacillus caldus* (GQ409763) | 83 | cbbM-OTU4 | *Magnetospirillum magnetotacticum* (AF442517) | 90 |
| cbbL-OTU5 | *Thiobacillus denitrificans* (L42940) | 86 | cbbM-OTU5 | *Acidithiobacillus ferrooxidans* (KT884974) | 98 |
| cbbL-OTU6 | *Acidithiobacillus ferrooxidans* (AB362170) | 100 | cbbM-OTU6 | Thiobacillus sp. ( AB032829) | 99 |
| cbbL-OTU7 | *Chromatium vinosum* (M26396) | 89 | | | |
| cbbL-OTU8 | *Thioalkalivibrio versutus* (KY452013) | 82 | | | |
| cbbL-OTU9 | *Azospirillum lipoferum* (DQ787337) | 100 | | | |

Table S2 The relative abundance of the main known groups in the heavy fractions of Yangshanchong mine tailings (YM) and Shuimuchong mine tailings (SM) according to DNA-SIP and 16S rRNA analysis.

| | YM | SM |
|---|---|---|
| k__Bacteria;p__Firmicutes;c__Clostridia;o__Clostridiales;f__Sulfobacillaceae;g__Sulfobacillus | 17.18% | 18.24% |
| k__Bacteria;p__Proteobacteria;c__Alphaproteobacteria;o__Sphingomonadales;f__Sphingomonadaceae;g__Novosphingobium | 2.87% | 3.40% |
| k__Bacteria;p__Proteobacteria;c__Alphaproteobacteria;o__Rhodospirillales;f__Acetobacteraceae;g__Acidiphilium | 3.05% | 3.24% |
| k__Bacteria;p__Nitrospirae;c__Nitrospira;o__Nitrospirales;f__[Leptospirillaceae];g__Leptospirillum | 1.32% | 1.58% |
| k__Bacteria;p__Firmicutes;c__Clostridia;o__Clostridiales;f__Sulfobacillaceae;g__YNPFFP6 | 1.14% | 1.20% |
| k__Bacteria;p__Firmicutes;c__Bacilli;o__Bacillales;f__Alicyclobacillaceae;g__Alicyclobacillus | 1.00% | 1.09% |
| k__Bacteria;p__Proteobacteria;c__Betaproteobacteria;o__Rhodocyclales;f__Rhodocyclaceae;g__Methyloversatilis | 0.87% | 1.00% |
| k__Bacteria;p__Actinobacteria;c__Thermoleophilia;o__Solirubrobacterales;f__Conexibacteraceae;g__Conexibacter | 0.39% | 0.43% |
| k__Bacteria;p__Proteobacteria;c__Gammaproteobacteria;o__Legionellales;f__Coxiellaceae;g__Aquicella | 0.25% | 0.38% |
| k__Archaea;p__Euryarchaeota;c__Thermoplasmata;o__Thermoplasmatales;f__Picrophilaceae;g__Thermogymnomonas | 0.23% | 0.15% |
| k__Bacteria;p__Proteobacteria;c__Alphaproteobacteria;o__Caulobacterales;f__Caulobacteraceae;g__Phenylobacterium | 0.14% | 0.15% |
| k__Bacteria;p__Proteobacteria;c__Alphaproteobacteria;o__Rhizobiales;f__Methylobacteriaceae;g__Methylobacterium | 0.13% | 0.13% |
| k__Bacteria;p__Proteobacteria;c__Betaproteobacteria;o__Rhodocyclales;f__Rhodocyclaceae;g__Dok59 | 0.03% | 0.11% |
| k__Bacteria;p__Proteobacteria;c__Gammaproteobacteria;o__Acidithiobacillales;f__Acidithiobacillaceae;g__Acidithiobacillus | 0.11% | 0.06% |

---

## Author Response (AR1)

**Point-by-point response to the comments of the reviewer #1**

1. **Comments from Referees**: The section of 2.1 should include more details about the soil sampling.

    **Response**: Thanks for this comment. I have rewritten the soil sampling details in the section of 2.1.

    **Changes in manuscript**:

    (Lines 69-71) "Samples of oxidized mine tailings on the surface (0-20 cm) were collected using a steel corer in October 2015. Mine tailing samples stored in sterilized plastic bags were transported to the laboratory in an ice cooler and stored at -20°C before analysis."

2. **Comments from Referees**: The experiment treatments was also not clear. From section 2.2, we can't judge how many treatments were set. Please specify. Moreover, what's samples were conducted qPCR, sequencing and cloning were also confused.

    **Response:** Thanks for this comment. I have rewritten the experiment treatments in sections. And samples that conducted qPCR, sequencing and cloning were also written. And the detail data for qPCR has been showed in the supplementary materials.

    **Changes in manuscript:**

    (1) For experiment treatment: (Lines 78-85) "There were a total of four treatments in the microcosms of the two mine tailings. For each mine tailing, fresh mine tailings (equivalent to 10.0 g d.w.s.) were mixed with a total of 2 g of sterile pulverized $FeS_2$ at approximately 60% maximum water-holding capacity as the $FeS_2$ treatment, followed by incubation at 25°C in the dark for 14 days. Yangshanchong mine tailing samples (YM) cultured with $FeS_2$ are abbreviated as YM_$FeS_2$, and Shuimuchong mine tailing samples (SM) cultured with $FeS_2$ are abbreviated as SM_$FeS_2$. In addition, fresh mine tailings at approximately 60% maximum water-holding capacity without any additive were used as the control groups and abbreviated as YM_ck and SM_ck. For each treatment, the microcosms were incubated with 10% $^{13}C$-$CO_2$ or $^{12}C$-$CO_2$, and both treatments were constructed in triplicate for DNA-SIP analysis."

    (2) For samples that conducted qPCR, sequencing and cloning: (Lines 110-112) "Real-time quantitative PCR analysis was performed on a CFX96 optical real-time detection system (Bio-Rad, Laboratories Inc., Hercules, CA, USA) to determine the copy numbers of the *cbbL*, *cbbM* and 16S rRNA genes in DNA gradient fractions from the YM_$FeS_2$ and SM_$FeS_2$ DNA-SIP microcosms."(Lines 126-129) "The composition of the bacterial communities in different samples was assessed by pyrosequencing of the 16S rRNA genes. The 16S rRNA gene from the $^{13}C$-labeled DNA fraction, which had CsCl buoyant densities of 1.738 g·mL$^{-1}$ in the heavy fraction in YM_$FeS_2$ and 1.734 g·mL$^{-1}$ in the heavy fraction in SM_$FeS_2$, was also amplified for pyrosequencing." (Lines 144-146) "Clone libraries of the *cbbL* and *cbbM* genes were also constructed from the $^{13}C$-labeled DNA fractions with CsCl buoyant densities of 1.738 g·mL$^{-1}$ in the heavy fraction in YM_$FeS_2$ and 1.734 g·mL$^{-1}$ in the heavy fraction in SM_$FeS_2$."

3. Comments from Referees: For section 4.1, authors only described how Carbon sequestration in mine tailings but without any discussion Combining with your own data.

    **Response:** Thanks for this comment. I have rewritten the discussion based on this comments.

**Changes in manuscript:**

(Lines 263-290) "4.1 The effect of FeS2 on the whole bacterial community in mine tailings

Acidic polymetallic mine tailings have strong potential for pyrite oxidation. In this study, a large amount of sulfuric acid was generated (increases of approximately 19.95 mg/g and 14.64 mg/g in YM and SM, respectively), and a persistent decline in pH was observed (pH decreased by 0.44 and 0.35 in YM and SM, respectively) in only 14 days. These changes clearly indicated oxidization of pyrite (i.e., acidification) in mine tailings. Previous studies have found that some bacterial phyla, such as Firmicutes and Nitrospirae, significantly increase (Chen et al., 2013;Liu et al., 2014) with the acidification process of mine tailings. In the present study, the bacterial composition in the different mine tailings varied greatly, and only the Firmicutes phylum increased in both tested mine tailings under pyrite addition. This group might participate in the oxidization of sulfide minerals (Chen et al., 2013), such as *Sulfobacillus*, which accounted for the majority of Firmicutes. Many other microorganisms might be inhibited under pyrite addition. Korehi et al. (2014) and Liu et al. (2014) also indicated that the ongoing oxidation process in mine tailings was accompanied by an increase in Firmicutes and a decrease in Actinobacteria and all classes of Proteobacteria except Gammaproteobacteria. In addition, Chen et al. (2013) and Liu et al. (2014) found that the relative abundances of Euryarchaeota belonging to archaea significantly increased with decreasing pH, which indicates that this taxon is an indicator of metal contamination (Hur et al., 2011). Euryarchaeota compete with β-Proteobacteria for ecological niches under such acidic conditions (Liu et al., 2014). However, in this study, only a few archaea were detected, which might be related to differences in primer affinities and samples.

The growth of microorganisms in bare mine tailings is usually limited by the availability of organic carbon (Schimel and Weintraub, 2003). Pyrite oxidization in mine tailings further enhanced the acidity of the mine tailings (pH decreased to 2.77 and 2.57 in YM_FeS$_2$ and SM_FeS$_2$, respectively). As a result, only microorganisms that were resistant to infertility and/or acidophilic conditions could maintain high activities. In the present study, *Conexibacter*, *Alicyclobacillus*, *Bacillus*, *Sulfobacillus*, *Leptospirillum*, *Rhodoplanes*, *Methylobacterium*, *Acidiphilium*, *Novosphingobium* and *Methyloversatilis* were the top genera with relative abundances above 0.2% in the mine tailings. Some specific taxa, including the genera *Alicyclobacillus*, *Sulfobacillus*, *Leptospirillum* and *Acidiphilium*, increased in both of the tested mine tailings under pyrite addition, indicating high consistency of dominant bacterial genera in different mine tailings. It is possible that in the case of pyrite oxidization and the availability of organic carbon, acidophilic and/or autotrophic bacteria could be stimulated (Deng et al., 2016;Antonelli et al., 2018), and the main carbon fixers found in these two mine tailings may be derived from the same groups."

**Point-by-point response to the comments of the reviewer #2**

1. **Comments from Referees**: The manuscript explored the autotrophic microbes and the FeS2 facilitation role in acidic mine tailings using stable isotope and molecular methods. The results showed that FeS2 facilitated $CO_2$-fixing by microbes and increased the abundances of relevant autotrophs. The study is very interesting, which could provide new insights into the autotrophic roles in extreme environments. However, the article writing is awful in logic, result description and interpretation.

**Response**: We thank the reviewer for pointing out the importance of this study and giving me a number of useful advices. And I make major revisions according to these comments.

2. **Comments from Referees**: The introduction did not show some key points relevant to the research, such as possible $CO_2$-fixing pathways and autotrophs in acidic sulfur-enriched environments. The introduction was not well structured and really needs rewrite.

**Response**: Thanks for this comment. I have rewritten the introduction details according to this comment.

**Changes in manuscript**:

[revised manuscript text omitted]

(7) Statistic software informations has been added. (Lines 154-159) "Bray-Curtis distance matrices for the overall bacterial community composition among the given samples were calculated in R v.3.3.2 using the 'vegdist' function of the vegan package and visualized by nonmetric multidimensional scaling (NMDS) in Origin 8. A heatmap of dominant genera with relative abundances above 0.2% was applied for plotting in the R environment with the pheatmap package. The translated cbbL and cbbM sequences from the heavy fractions were used to construct a phylogenetic tree by the neighbor-joining method using the MEGA package, version 7.0."

(8) I have rewritten the experiment treatments in sections. (Lines 175-182) "There were a total of four treatments in the microcosms of the two mine tailings. For each mine tailing, fresh mine tailings (equivalent to 10.0 g d.w.s.) were mixed with a total of 2 g of sterile pulverized FeS2 at approximately 60% maximum water-holding capacity as the FeS2 treatment, followed by incubation at 25°C in the dark for 14 days. Yangshanchong mine tailing samples (YM) cultured with FeS2 are abbreviated as YM_FeS2, and Shuimuchong mine tailing samples (SM) cultured with FeS2 are abbreviated as SM_FeS2. In addition, fresh mine tailings at approximately 60% maximum water-holding capacity without any additive were used as the control groups and abbreviated as YM_ck and SM_ck. For each treatment, the microcosms were incubated with 10% 13C-CO2 or 12C-CO2, and both treatments were constructed in triplicate for DNA-SIP analysis."

**4. Comments from Referees**: Fig1 symbols are very confusing, and no descriptions on the above and bottom columns.

**Response**: Thanks for this comment. I have redrawn the figure and have rewritten the figure legend.

**Changes in manuscript**:

[Figure]

Fig. 1 pH values (a), 13C atom % (b), TOC (c) contents, SO42- (d), Fe2+ (e) and Fe3+ (f) in mine tailings. The error bars indicate the standard errors of three subsamples for each tailing sample. To determine 13C atom % (b), all analyzed samples were treated with 13C-CO2 in microcosms. YM_ck, control group of Yangshanchong mine tailings; SM_ck, control group of Shuimuchong mine tailings; YM_FeS2, Yangshanchong mine tailings treated with FeS2; SM_FeS2, Shuimuchong mine tailings treated with FeS2.

**5. Comments from Referees**: No specific legends or descriptions on the two inserts in Fig. 3, and the color differences are not clear

**Response**: Thanks for this comment. I have redrawn the figure and have rewritten the figure legend.

**Changes in manuscript**:

[Figure]

Fig. 3 Relative abundances (percentages) of the main identified bacterial taxonomic groups, i.e.,

the phyla Proteobacteria, Firmicutes, Acidobacteria, Actinobacteria, WPS-2, Planctomycetes, AD3 and Nitrospirae (a); and the classes Alphaproteobacteria, Betaproteobacteria, Deltaproteobacteria and Gammaproteobacteria (within the phylum Proteobacteria) (b). For each tailing sample, the relative abundances of the sequences assigned to a given taxonomic unit were calculated for each of three subsamples, and the average value was then used to represent the relative abundance of each tailing sample.

**6. Comments from Referees**: L252-261, 284-291, there are many super long sentences. A sentence usually contains maximum 22 words.

**Response**: Thanks for this comment. I have rewritten these sentences.

**Changes in manuscript**:

(1) (Lines 193-200) "In the Yangshanchong mine tailings, pyrite addition significantly increased the relative abundances of AD3, Nitrospirae and unclassified Proteobacteria by 0.75% (P=0.008), 0.59% (P=0.019) and 6.33% (P<0.001), respectively. In parallel, in Shuimuchong mine tailings, FeS2 addition significantly increased the relative abundances of Firmicutes, Planctomycetes, unclassified Proteobacteria, Alphaproteobacteria, Betaproteobacteria, and Deltaproteobacteria by 15.69% (P<0.001), 0.97% (P<0.001), 5.88% (P=0.002), 4.35% (P=0.001), 8.61% (P<0.001) and 0.21% (P=0.003), respectively. However, in SM, the percentages of AD3, Acidobacteria, Actinobacteria and Gammaproteobacteria in SM by 0.97% (P=0.002), 7.43% (P=0.002), 1.35% (P=0.016) and 4.85% (P=0.002) were decreased under pyrite addition."

(2) (Lines 217-223) "In the Yangshanchong mine tailings, pyrite addition significantly increased the relative abundances of the genera Alicyclobacillus, Leptospirillum, Sulfobacillus and Acidiphilium by 0.67% (P=0.027), 0.74% (P=0.002), 8.86% (P=0.043) and 1.57% (P<0.001), respectively. In the Shuimuchong mine tailings, FeS2 addition significantly increased the relative abundances of Alicyclobacillus, Bacillus, Sulfobacillus, Methylobacterium, Novosphingobium and Methyloversatilis by 0.57% (P<0.001), 9.94% (P<0.001), 5.99% (P<0.001), 0.15% (P<0.001), 2.06% (P=0.004) and 0.56% (P<0.001), respectively."

**7. Comments from Referees**: The $CO_2$-fixing capacity by autotrophs should be calculated.

**Response**: Thanks for this comment. I have calculated the $CO_2$-fixing capacity by autotrophs in the result.

**Changes in manuscript**:

(Lines 173-174) "The CO2-fixing capacities of autotrophs under $FeS_2$ addition were 9.50±0.91 mg/kg·d in YM and 3.69±0.11 mg/kg·d in SM."

**8. Comments from Referees**: L306-307, the statement is problematic: 12CO2 is a control relative to 13CO2, so the shift to heavy fractions should not be observed in 12CO2. L307-311, the statements are not correct: for the peak in 13CO2 occurred in the density of 1.72 rather than 1.73 in both 12CO2 and 13CO2.

**Response**: Thanks for this comment. I have written the sentences.

**Changes in manuscript**:

(Lines 235-238) "*cbbL* and *cbbM* gene levels under $^{13}C$-$CO_2$ treatment peaked at a density of 1.72 in both the $^{12}C$-$CO_2$ and $^{13}C$-$CO_2$ treatments. In addition, a shift toward heavy fractions was observed for the *cbbL* and *cbbM* gene abundances in the $^{13}C$-$CO_2$ treatment (Fig. 5), with buoyant

densities of 1.738 g·mL$^{-1}$ in YM_FeS2 and 1.734 g·mL-1 in SM_FeS2."

**9. Comments from Referees**: L311-314, the statements should go to discussion section
**Response**: Thanks for this comment. I have move the works into the discussion section.
**Changes in manuscript**:
(Lines 305-309) "In addition, only a few archaea were detected based on 16S rRNA gene sequencing, and the clone libraries of the *cbbL* and *cbbM* genes in the 13C-labeled heavy fraction did not show archaeal sequences for Calvin cycle genes. These results indicated that bacterial carbon sequestration is mainly attributable to chemoautotrophic bacteria in pyrite oxidization of mine tailings. However, archaea may have higher activities in RuBisCO-mediated carbon metabolic pathways (Kono et al., 2017), which will require further study."

**10. Comments from Referees**: Fig. 6, Cultured genus most related to OTU1, 2, 3 and 4 should be given for identifying purpose
**Response**: Thanks for this comment. I have added more information about the cultured genus most related to OTU1, 2, 3 and 4 in the supply materials
**Changes in manuscript**:
**Fig. 6** Phylogenetic tree of the translated *cbbL* and *cbbM* sequences in the heavy fractions from YM and SM treated with FeS$_2$. Relative frequencies (%) are marked in the bar graph. Bootstrap values of >50% are indicated at branch points. The *cbbL* and *cbbM* gene copy numbers in the heavy fractions from FeS$_2$-treated YM and SM are shown in the middle of the figure. The cultured genera most related to OTUs from the *cbbL* and *cbbM* clone libraries are shown in Table S1.

Table S1 Cultured genus most related to OTUs from *cbbL* and *cbbM* clone libraries

| cbbL OTU | Closest affiliation | Max identity | cbbM OTU | Closest affiliation | Max identity |
|---|---|---|---|---|---|
| cbbL-OTU1 | *Sulfobacillus acidophilus* (DQ139404) | 99 | cbbM-OTU1 | Gamma proteobacterium (HQ675075) | 98 |
| cbbL-OTU2 | *Thiohalospira halophila* (GQ888600) | 83 | cbbM-OTU2 | *Magnetospirillum moscoviense* (KF712469) | 88 |
| cbbL-OTU3 | *Chromatium vinosum* (M26396) | 78 | cbbM-OTU3 | Gamma proteobacterium (HQ675084) | 98 |
| cbbL-OTU4 | *Acidithiobacillus caldus* (GQ409763) | 83 | cbbM-OTU4 | *Magnetospirillum magnetotacticum* (AF442517) | 90 |
| cbbL-OTU5 | *Thiobacillus denitrificans* (L42940) | 86 | cbbM-OTU5 | *Acidithiobacillus ferrooxidans* (KT884974) | 98 |
| cbbL-OTU6 | *Acidithiobacillus ferrooxidans* (AB362170) | 100 | cbbM-OTU6 | Thiobacillus sp. ( AB032829) | 99 |
| cbbL-OTU7 | *Chromatium vinosum* (M26396) | 89 | | | |
| cbbL-OTU8 | *Thioalkalivibrio versutus* (KY452013) | 82 | | | |
| cbbL-OTU9 | *Azospirillum lipoferum* (DQ787337) | 100 | | | |

**11. Comments from Referees**: cbb is not a correct gene name, it should be cbbL or cbbM.
**Response**: Thanks for this comment. I have rewritten the gene name across the manuscript.
**Changes in manuscript**:

 "The *cbbL* and *cbbM* gene sequences from the clone libraries in the $^{13}$C-DNA heavy fraction treated with $^{13}$C-CO$_2$ were used for phylogenetic analysis (Fig. 6)."
 "For the quantitative analysis of *cbbL* and *cbbM* gene abundances, the buoyant densities of the DNA in isopycnic centrifugation gradients were used to assess the labeling efficiencies of *cbbL* or *cbbM* gene-carrying carbon fixers in the DNA-SIP microcosms (Fig. 5)."

**12. Comments from Referees**: Is Fig. 5 for FeS2 treatments or raw mine tailings?
**Response**: Thanks for this comment. The figure was for FeS2 treatment and I have redrawn the figure. And *cbbL* and *cbbM* gene abundance qPCR data are shown in the supplementary materials.
**Changes in manuscript**:

[Figure]

**Fig. 5** Quantitative distribution of *cbbL* and *cbbM* gene fragments across the entire buoyant density gradients of the DNA fractions from microcosms treated with FeS$_2$ and incubated with $^{12}$C-CO$_2$ or $^{13}$C-CO$_2$. The normalized data consist of the ratio of the gene copy number for each DNA gradient to the maximum quantity for each treatment. The error bars represent the standard errors of triplicate microcosms, and each contains three technical replicates. *cbbL* and *cbbM* gene abundance qPCR data are shown in the supplementary materials.

**13. Comments from Referees**: L351-371, the paragraph should go to introduction section. The discussion is far from the results, i.e. discuss why and how FeS2 facilitates microbial CO2-fixing and changes the whole bacterial community.
**Response:** Thanks for this comment. I have rewritten the Introduction and Discussion according to this comment.
**Changes in manuscript**:

[revised manuscript text omitted]

---

## Author Response (AR2)

**Point-by-point response to the comments of the reviewer #2**

1. **Comments from Referees:** I know it is a revised manuscript, which was substantially improved. However, I still have some concerns. The results between bacterial community structure changes and carbon fixing bacteria were disconnected. Furthermore, the manuscript is written with plenty of grammatical mistakes, therefore I suggest authors to seek assistance from a native speaker for proofreading.

   **Response**: We thank the reviewer for giving me a number of useful advices. And I make major revisions according to these comments. Firstly, the dominant genera analysis from results section 3.2 was rewritten, which pointed out the dominant genera in the pyrite oxidation of mine tailings and also highlighted the importance of the autotrophic genera in samples. Similarly, the fig.4 has been redrawn to reflect the dominant genera including autotrophic taxon in mine tailings. Secondly, the English Language is revised by the highly qualified native English speaking editors at American Journal Experts after we rewrote the manuscript.

[Figure]

   **Changes in the result 3.2 from manuscript**:

   (Lines 207-221) "The total number of genera assigned to known taxa accounted for 29.89% of the total bacterial communities. In addition, we constructed a heatmap diagram (Fig. 4) that shows the top 51 dominant genera with relative abundances above 0.02% in the mine tailings, accounting for 29.16% of the total bacterial communities. Specifically, *Sulfobacillus* (8.04%) and *Novosphingobium* (8.60%) accounted for 16.64% of the total bacterial communities and were the dominant taxa in the mine tailings. In contrast, autotrophic bacteria including *Rhodanobacter* (0.04%), *Pseudomonas* (0.02%), *Acidithiobacillus* (0.02%), *Thiobacillus* (0.04%), *Ralstonia* (0.02%), *Thiomonas* (0.04%), *Burkholderia* (0.09%), *Acidiphilium* (1.49%), *Rhodobacter* (0.04%), *Rhodoplanes* (0.59%), *Nitrospira* (0.02%), *Leptospirillum* (0.80%), *Sulfobacillus* (8.04%), *Clostridium* (0.04%) and *Corynebacterium* (0.04%) accounted for 11.33% of the total bacterial communities. Whereby, *Thiobacillus*, *Acidiphilium*, *Leptospirillum*, *Acidithiobacillus* and *Sulfobacillus* are ferrous and sulfur-oxidizing bacteria. For the Yangshanchong mine tailings, pyrite addition significantly increased the relative abundances of the autotrophic genera *Acidithiobacillus*, *Leptospirillum*, *Sulfobacillus* and *Acidiphilium* by 0.02% ($P$=0.001), 0.74%

($P$=0.002), 8.86% ($P$=0.043) and 1.57% ($P$<0.001), respectively. $FeS_2$ addition also significantly increased the relative abundances of autotrophic genera in the Shuimuchong mine tailings: *Rhodanobacter*, *Acidithiobacillus*, *Thiobacillus* and *Sulfobacillus* by 0.07% ($P$=0.016), 0.03% ($P$=0.034), 0.02% ($P$=0.030) and 5.99% ($P$<0.001), respectively."

2. **Comments from Referees:** line 30 'chemolithoautotrophic organisms fix atmospheric CO2 by six pathways' this sentence is not connected to the previous sentence, the last sentence introduced plant photosynthesis, then it jumped into cehmolithoautotrophs. Furtheremore, there was no mention of autotrophic bacteria which also use cbb cycle.
   **Response**: Thanks for this comment. I rewrote the first paragraph of the introduction to improve the logicality of the article, and I also pointed out that "the CBB cycle is the most prevalent means of $CO_2$ fixation by autotrophs including autotrophic bacteria" in Lines 34-35.
   **Changes in manuscript**:
   (Lines 28-38) "Soil ecosystems have great potential as carbon sinks to stabilize $CO_2$ and regulate climate change (White et al., 2000). Atmospheric $CO_2$ can be fixed in plants via photosynthesis and assimilated into soils via decomposition and microbial activity (Deng et al., 2016;Antonelli et al., 2018), and autotrophic bacteria play a significant role in carbon sequestration in soil ecosystems (Berg, 2011;Alfreider et al., 2017). Six autotrophic carbon sequestration mechanisms are widespread, including the Calvin-Benson-Bassham (CBB) cycle, the reductive tricarboxylic acid (rTCA) cycle, the reductive acetyl-CoA pathway and the recently discovered 3-hydroxypropionate/4-hydroxybutyrate (HP/HB) cycles (Berg, 2011;Alfreider et al., 2017). Among them, the CBB cycle is the most prevalent means of $CO_2$ fixation by autotrophs including autotrophic bacteria (Tabita, 1999;Berg, 2011). The enzyme ribulose-1,5-bisphosphate carboxylase/oxygenase (RuBisCO) is important in the CBB cycle and is in fact the most prominent enzyme on Earth (Raven, 2013). The *cbbL* and *cbbM* genes encoding the large subunit of RuBisCO, with 25 to 30% amino acid sequence identity (Tabita et al., 2008), serve as autotroph markers (Berg, 2011;Alfreider et al., 2017)."

3. **Comments from Referees:** line 35-40 The *cbbL* and *cbbM* genes encode RuBisCO form I and form II, this is incorrect, they encode the large submit of RuBisCO
   **Response**: Thanks for this comment and point out mistakes in my limited knowledge. And I have written the sentences.
   **Changes in manuscript**:
   (Lines 37-38) "The *cbbL* and *cbbM* genes encoding the large subunit of RuBisCO, with 25 to 30% amino acid sequence identity (Tabita et al., 2008), serve as autotroph markers (Berg, 2011;Alfreider et al., 2017)."

4. **Comments from Referees:** line 80 For 13CO2 labelling experiment, how the 13CO2 were delievered?
   **Response**: Thanks for this comment. And I have written the sentences.
   **Changes in manuscript**:
   (Lines 78-80) "The microcosms were incubated with 10% $^{13}C$-$CO_2$ or $^{12}C$-$CO_2$, and both

treatments were constructed in triplicate for DNA-SIP analysis."

5. **Comments from Referees:** line 170-175 'Furthermore, the 13C atom % values in YM_FeS2 and SM_FeS2 were higher than those in the control groups YM_ck and SM_ck, which exhibited 13C atom % values of 1.76±0.06 and 1.36±0.01.' there were four sample groups (YM_FeS2, SM_FeS2, YM, and SM, both with 13C labelling and without) but only two 13C% were provided.
   **Response**: Thanks for this comment. And I have written the sentences.
   **Changes in manuscript**:
   (Lines 165-167) "and the $^{13}C$ atom % values in YM_FeS$_2$ (1.76±0.06 $^{13}C$ atom %) and SM_FeS$_2$ (1.76±0.06 $^{13}C$ atom %) were higher than those in the controls YM_ck (1.12±0.01 $^{13}C$ atom %) and SM_ck (1.11±0.01 $^{13}C$ atom %)."

6. **Comments from Referees:** line 190-195 'a total of 8 bacterial phyla and 4 proteobacterial classes were frequently identified in the two mine tailings' please define frequently
   **Response**: Thanks for this comment. And I have written the sentences.
   **Changes in manuscript**:
   (Lines 185-186) "In this study, 8 dominant bacterial phyla/candidate divisions (relative abundance >1%) and 4 proteobacterial classes were identified in the two mine tailings,"

7. **Comments from Referees:** line 210-215 '...that shows the top 39 dominant genera with relative abundances above 0.20% from the mine tailings.' some of these taxa are not at genus level (i.e., all those unclassified taxa), they are sequences unclassified at various taxonomic levels.
   line 210-215 'only a small number of genera were assigned to known taxa' If they can't be classified at genus level, how can author be certain that they are indeed bacterial genus?
   Fig.4 there are some taxonomic units that presented at very high relative abundances (such as unclassified Commamonadaceae, Unclassified Xanthomonadaceae, Novosphingobium, Unclassified proteobacteria), which were much higher than the dominant carbon fixers such as Sulfobacillum, Leptospirillum. Is there any explaination for this?
   **Response**: Thanks for these comments. And I have written the results section 3.2 which pointed out the dominant genera in the pyrite oxidation of mine tailings and also highlighted the importance of the autotrophic genera in samples. Those unclassified taxa have been removed, and the Fig.4 has been redrawn.
   **Changes in manuscript**:
   (Lines 207-221) "The total number of genera assigned to known taxa accounted for 29.89% of the total bacterial communities. In addition, we constructed a heatmap diagram (Fig. 4) that shows the top 51 dominant genera with relative abundances above 0.02% in the mine tailings, accounting for 29.16% of the total bacterial communities. Specifically, *Sulfobacillus* (8.04%) and *Novosphingobium* (8.60%) accounted for 16.64% of the total bacterial communities and were the dominant taxa in the mine tailings. In contrast, autotrophic bacteria including *Rhodanobacter* (0.04%), *Pseudomonas* (0.02%),

*Acidithiobacillus* (0.02%), *Thiobacillus* (0.04%), *Ralstonia* (0.02%), *Thiomonas* (0.04%), *Burkholderia* (0.09%), *Acidiphilium* (1.49%), *Rhodobacter* (0.04%), *Rhodoplanes* (0.59%), *Nitrospira* (0.02%), *Leptospirillum* (0.80%), *Sulfobacillus* (8.04%), *Clostridium* (0.04%) and *Corynebacterium* (0.04%) accounted for 11.33% of the total bacterial communities. Whereby, *Thiobacillus*, *Acidiphilium*, *Leptospirillum*, *Acidithiobacillus* and *Sulfobacillus* are ferrous and sulfur-oxidizing bacteria. For the Yangshanchong mine tailings, pyrite addition significantly increased the relative abundances of the autotrophic genera *Acidithiobacillus*, *Leptospirillum*, *Sulfobacillus* and *Acidiphilium* by 0.02% ($P$=0.001), 0.74% ($P$=0.002), 8.86% ($P$=0.043) and 1.57% ($P$<0.001), respectively. $FeS_2$ addition also significantly increased the relative abundances of autotrophic genera in the Shuimuchong mine tailings: *Rhodanobacter*, *Acidithiobacillus*, *Thiobacillus* and *Sulfobacillus* by 0.07% ($P$=0.016), 0.03% ($P$=0.034), 0.02% ($P$=0.030) and 5.99% ($P$<0.001), respectively."

[Figure]

**Fig. 4** Heatmap of the top genera with relative abundances above 0.02% in mine tailings. Autotrophic bacteria were marked with underlining.

8. **Comments from Referees:** line 265-270 'a large amount of sulfuric acid was generated (increases of approximately 19.95 mg/g and 14.64 mg/g in YM and SM, respectively), and a persistent decline in pH was observed (pH decreased by 0.44 and 0.35 in YM and SM, respectively) in only 14 days.' Inconsistent with results, results paragraph 1 stated no significant changes in pH

   **Response**: Thanks for this comment. Due to the unclear statement in previous paper version, reviewers and readers do not clear what I mean. In the result, I mean that no significant changes in chemical properties were found for the control groups compared to the original Yangshanchong and Shuimuchong acidic samples, it was pointed out the chemical properties in the YM_ck and SM_ck samples compared to the original samples showed in Lines 69-73 ("The properties of the mine tailings were as follows: Yangshanchong acidic samples, pH 3.21, total nitrogen (TN) 0.11 $g·kg^{-1}$, total organic carbon (TOC) 16 $g·kg^{-1}$, $SO_4^{2-}$ 13.32 $g·kg^{-1}$, $As_T$ 63.29 $mg·kg^{-1}$, $Fe_T$ 133.46 $g·kg^{-1}$, $Cu_T$ 1.95 $g·kg^{-1}$, $Pb_T$ 27.58 $mg·kg^{-1}$, and $Zn_T$ 205.44 $mg·kg^{-1}$; Shuimuchong acidic samples, pH 2.92, TN 0.11 $g·kg^{-1}$, TOC 18 $g·kg^{-1}$, $SO_4^{2-}$ 8.84 $g·kg^{-1}$, $As_T$ 51.77 $mg·kg^{-1}$, $Fe_T$ 117.59 $g·kg^{-1}$, $Cu_T$ 2.53 $g·kg^{-1}$, $Pb_T$ 30.43 $mg·kg^{-1}$, and $Zn_T$ 176.59 $mg·kg^{-1}$."). And pyrite addition decreased pH values in the YM and SM samples by 0.48±0.16 and 0.41±0.07, respectively. Pyrite addition also increased the $SO_4^{2-}$ content by 252.96% and 262.35%, $Fe^{2+}$ content by 329.47% and 240.38%, and $Fe^{3+}$ content by 137.47% and 140.37% in the YM and SM samples, respectively.

   **Changes in manuscript**:

   (Lines 157-164) "No significant changes in chemical properties, pH values (3.25±0.09 in YM_ck and 2.98±0.04 in SM_ck), sulfate ($SO_4^{2-}$) contents (13.15±2.58 mg/g in YM_ck and 8.95±2.19 mg/g in SM_ck), and TOC contents (16.75±0.09 mg/g in YM_ck and 18.55±0.12 mg/g in SM_ck), were found for the control groups compared to the original Yangshanchong and Shuimuchong acidic samples after 14 days of incubation (Fig. 1). However, the addition of pyrite decreased pH values in the YM and SM samples by 0.48±0.16 and 0.41±0.07, respectively. Pyrite addition also increased the $SO_4^{2-}$ content by 252.96% and 262.35%, $Fe^{2+}$ content by 329.47% and 240.38%, and $Fe^{3+}$ content by 137.47% and 140.37% in the YM and SM samples, respectively. Together, these data indicate the occurrence of pyrite oxidization and acidification in mine tailings after pyrite addition."

9. **Comments from Referees:** line 285-290 'Some specific taxa, including the genera Alicyclobacillus, Sulfobacillus, Leptospirillum and Acidiphilium, increased in both of the tested mine tailings under pyrite addition' which figure does this statement refer to? Fig 4? please specify

   **Response**: Thanks for this comment. And I have written the sentences.

   **Changes in manuscript**:

   (Lines 280-283) "The level of some specific taxa, including the autotrophic genera *Acidithiobacillus* and *Sulfobacillus* increased in both of the tested mine tailings under pyrite addition (Fig.4), indicating high consistency of dominant autotrophic bacterial genera in different mine tailings."

10. **Comments from Referees:** line 295-300 the 13C content and TOC content in mine tailings

increased slightly which figure/table does this result refer to fig 1? please specify

**Response**: Thanks for this comment. And I have written the sentences.

**Changes in manuscript**:

(Lines 293-294) "However, in this study, the $^{13}$C and TOC contents in mine tailings increased slightly (Fig.1)."

11. **Comments from Referees:** line 295 to 300 'DNA-SIP analysis demonstrated that a considerable amount of 13C-CO2 was assimilated by carbon fixers in the 13C-CO2-labeled mine tailing samples, leading to a significant shift of cbbL or cbbM gene-carrying genomic DNA into the heavy fraction.' There was a new peak generated at Buoyant density of ~1.72, with density lower than the peak in control experiment, any explaination for the appearance of this peak? To me it is like a large proportion of autotrophic micoorganisms detected in 13C sample did not fix any carbon, this might explain why the intensity of 13C peak at higher density (1.735-1.74) was much lower compared with 12C peak.

**Response**: Thanks for this comment. And I have written the sentences.

**Changes in manuscript**:

(Lines 296-300) "In addition, a peak at a buoyant density of 1.72 g·mL$^{-1}$ in $^{13}$C-CO$_2$-labeled mine tailing samples was observed, with a density lower than the peak in the $^{12}$C-CO$_2$-labeled control experiment (see supply material.xlsx); the intensity of the $^{13}$C peak at a higher density of 1.738 g·mL$^{-1}$ was also much lower than the $^{12}$C peak at a higher density of 1.72 g·mL$^{-1}$. This suggests that a large proportion of the autotrophic microorganisms detected in the mine tailings samples did not fix carbon."

12. **Comments from Referees:** line 305 'in addition, only a few archaea were detected based on 16S rRNA gene sequencing, and the clone libraries of the cbbL and cbbM genes in the 13C-labeled heavy fraction did not show archaeal sequences for Calvin cycle genes.i does the primer used in this study target Archaea sequence?

**Response**: Thanks for this comment. And I have written the sentences.

**Changes in manuscript**:

(Lines 308-309) "Nonetheless, archaea may have higher activities in RuBisCO-mediated carbon metabolic pathways (Kono et al., 2017), which will require further study."

13. **Comments from Referees:** line 315-320 '...and the glycolic acid in all of these acidophiles might be due to the activity of RuBisCO' RuBisCO produce glycerate-3-phosphate, not glycolic acid

**Response**: Thanks for this comment. And I have written the sentences to correct my understanding.

**Changes in manuscript**:

(Lines 314-317) "among acidophilic prokaryotes isolated from mine-impacted environments, the ability to metabolize glycerate-3-phosphate appeared to be restricted to Firmicutes (e.g., *Sulfobacillus*) and that the glycerate-3-phosphate present in all of these acidophiles might be due to the activity of RuBisCO."

14. **Comments from Referees:** line 320-325 'None of the cbbL or cbbM genes identified were

highly homologous to genes in Leptospirillum, but this may be due to primer specificity' later author mentioned that Leptospirillum only had rTCA, then there is no suprise tha no cbbL gene classified as Leptospirillum been identified, and this has nothing to do with primer specificity.

**Response**: Thanks for this comment. And I have written the sentences.

**Changes in manuscript**:

(Lines 321-327) "Although none of the *cbbL* or *cbbM* genes identified in our study were highly homologous to genes in *Leptospirillum*, Marín et al. (2017) reported that rTCA carbon fixation pathway genes were mainly found in *Leptospirillum* spp. RuBisCO is the most prominent enzyme, and the genes encoding the large subunit of RuBisCO serve as a marker for the analysis of autotrophic organisms, including bacteria, using the CBB cycle (Berg, 2011). The *Sulfobacillus*-like *cbbL* gene dominated the 13C-labeled DNA among carbon-fixing taxa, and the higher relative abundance of *Sulfobacillus* than *Leptospirillum*, according to 16S rRNA analysis, demonstrates the contribution of the *Sulfobacillus*-like *cbbL* gene to carbon sequestration."

15. **Comments from Referees:** line 335-340 'Our results demonstrated higher 13C atom % values with the addition of pyrite than in control groups after a 14-day incubation, as well as a significant increase in the total organic carbon content.' In the results part author said "total organic carbon contents (16.75±0.09 mg/g in YM_ck and 18.55±0.12 mg/g in SM_ck) exhibited no significant changes after 14 days", so was there significant changes in TOC or not?

**Response**: Thanks for this comment. Due to the unclear statement in previous paper version, reviewers and readers do not clear what I mean. In the result, I mean that no significant changes in chemical properties were found for the control groups compared to the original Yangshanchong and Shuimuchong acidic samples, it was pointed out the chemical properties in the YM_ck and SM_ck samples compared to the original samples showed in Lines 69-73 ("The properties of the mine tailings were as follows: Yangshanchong acidic samples, pH 3.21, total nitrogen (TN) 0.11 $g·kg^{-1}$, total organic carbon (TOC) 16 $g·kg^{-1}$, $SO_4^{2-}$ 13.32 $g·kg^{-1}$, $As_T$ 63.29 $mg·kg^{-1}$, $Fe_T$ 133.46 $g·kg^{-1}$, $Cu_T$ 1.95 $g·kg^{-1}$, $Pb_T$ 27.58 $mg·kg^{-1}$, and $Zn_T$ 205.44 $mg·kg^{-1}$; Shuimuchong acidic samples, pH 2.92, TN 0.11 $g·kg^{-1}$, TOC 18 $g·kg^{-1}$, $SO_4^{2-}$ 8.84 $g·kg^{-1}$, $As_T$ 51.77 $mg·kg^{-1}$, $Fe_T$ 117.59 $g·kg^{-1}$, $Cu_T$ 2.53 $g·kg^{-1}$, $Pb_T$ 30.43 $mg·kg^{-1}$, and $Zn_T$ 176.59 $mg·kg^{-1}$."). "Additionally, the TOC content increased by 0.20±0.11 mg/g in $YM\_FeS_2$ and 0.28±0.14 mg/g in $SM\_FeS_2$, and the $^{13}C$ atom % values in $YM\_FeS_2$ (1.76±0.06 $^{13}C$ atom %) and $SM\_FeS_2$ (1.76±0.06 $^{13}C$ atom %) were higher than those in the controls YM_ck (1.12±0.01 $^{13}C$ atom %) and SM_ck (1.11±0.01 $^{13}C$ atom %). This result shows that fixation of $^{13}C$-$CO_2$ occurred in these mine tailings with the addition of pyrite; the $CO_2$-fixing capacities of autotrophs under $FeS_2$ addition were 9.50±0.91 mg/kg·d in YM and 3.69±0.11 mg/kg·d in SM."

**Changes in manuscript**:

(Lines 157-160) "No significant changes in chemical properties, pH values (3.25±0.09 in YM_ck and 2.98±0.04 in SM_ck), sulfate ($SO_4^{2-}$) contents (13.15±2.58 mg/g in YM_ck and 8.95±2.19 mg/g in SM_ck), and TOC contents (16.75±0.09 mg/g in YM_ck and 18.55±0.12 mg/g in SM_ck), were found for the control groups compared to the original

Yangshanchong and Shuimuchong acidic samples after 14 days of incubation (Fig. 1)."

(Lines 164-169) "Additionally, the TOC content increased by 0.20±0.11 mg/g in YM_FeS$_2$ and 0.28±0.14 mg/g in SM_FeS$_2$, and the $^{13}$C atom % values in YM_FeS$_2$ (1.76±0.06 $^{13}$C atom %) and SM_FeS$_2$ (1.76±0.06 $^{13}$C atom %) were higher than those in the controls YM_ck (1.12±0.01 $^{13}$C atom %) and SM_ck (1.11±0.01 $^{13}$C atom %). This result shows that fixation of $^{13}$C-CO$_2$ occurred in these mine tailings with the addition of pyrite; the CO$_2$-fixing capacities of autotrophs under FeS$_2$ addition were 9.50±0.91 mg/kg·d in YM and 3.69±0.11 mg/kg·d in SM."

---

## Author Response (AR3)

**Point-by-point response to the comments of Editor**

1. **Comments:** Line 14: Add % after 1.76±0.06;
   **Response**: Thanks for this comment and point out my mistakes.
   **Changes in manuscript**:
   (Line 14) "1.76±0.06% for Yangshanchong and 1.36±0.01% for Shuimuchong".

2. **Comments:** Line 12: Repetitive: 13C-labeled CO2 was evaluated using 13C isotope;
   **Response**: Thanks for this comment and point out my mistakes. And the sentence has been rewritten.
   **Changes in manuscript**:
   (Lines 11-13) "In this study, carbon sequestration in two samples of mine tailings treated with FeS$_2$ was evaluated using $^{13}$C isotope, pyrosequencing and DNA-based stable isotope probing (SIP) analyses to identify carbon fixers."

3. **Comments:** Line 18 Sulfobacillus (8.04%) and Novosphingobium(8.60%) dominating in which site? And changes from xxx?
   **Response**: Thanks for this comment and point out my mistakes. And the sentence has been readjustment and rewritten.
   **Changes in manuscript**:
   (Lines 15-18) "which emphasized the role of autotrophs in carbon sequestration with pyrite addition. Pyrite treatment also led to changes in the composition of bacterial communities, and several autotrophic bacteria increased including *Acidithiobacillus* and *Sulfobacillus*. And pyrite addition increased the relative abundance of dominant genus *Sulfobacillus* by 8.86% and 5.99% in Yangshanchong and Shuimuchong samples, respectively."

4. **Comments:** Line 175 you can not say "duringthe pyrite oxidation process", as the samples were collected at one time point.
   **Response**: Thanks for this comment and point out my mistakes. And the sentence has been readjustment and rewritten.
   **Changes in manuscript**:
   (Line 177) "3.2 Bacterial communities in mine tailings under FeS2 addition"

5. **Comments:** Fig. 1b 13C should be $^{13}$C
   **Response**: Thanks for this comment and point out my mistakes. And figure has been redrawn.
   **Changes in manuscript**:

[Figure]

6.  **Comments:** In the end, I still doubt TOC can be increased as 0.20±0.11mg/g in just 14 days. Please give some support (lieteratures, proposed mechanisms, etc) in the section of "Discussion".

    **Response:** Thanks for this comment and point out the doubt. And I also confused by this data. In fact, the change of this data is very little. And both the soil acidification pretreatment before analysis and the addition of 20% $FeS_2$ in samples could increase the error. Therein, for the TOC in YM_FeS$_2$ and SM_FeS$_2$, the soil quality should be convert according to the addition of 20% $FeS_2$, which also could overestimate the increment of TOC. Thus this data should not be an absolute proof, and I also rewrite the sentence of increased TOC in abstract and conclusion. And the discussion also pointed out the opinion. Furthermore, I think the strict TOC increment in the acidification process of mine tailings must be used a long-term experiment.

    **Changes in manuscript**:

    Abstract: (Lines 13-15) "Mine tailings treated with $FeS_2$ exhibited a higher percentage of $^{13}C$ atoms (1.76±0.06% for Yangshanchong and 1.36±0.01% for Shuimuchong) than did controls over a 14-day incubation, which emphasized the role of autotrophs in carbon sequestration with pyrite addition."

    Conclusion: (Lines 341-342) "Our results reveal higher $^{13}C$ atom % values with the addition of pyrite than in controls after a 14-day incubation."

    Discussion: (Lines 293-297) "Although the results showed that TOC content in mine tailings increased slightly under $FeS_2$ addition, the soil acidification pretreatment and the addition of 20% $FeS_2$ in samples could increase the error of TOC analysis and calculation. And a long-term field test should be used to calculate the TOC increment in the acidification process of mine tailings in the future. Even so, in this study, the $^{13}C$ content in mine tailings increased significantly."